# WorldCoder, a Model-Based LLM Agent: Building World Models by Writing Code and Interacting with the Environment

**Hao Tang**
Cornell University
haotang@cs.cornell.edu

**Darren Key**
Cornell University
dyk34@cornell.edu

**Kevin Ellis**
Cornell University
kellis@cornell.edu

## Abstract

We give a model-based agent that builds a Python program representing its knowledge of the world based on its interactions with the environment. The world model tries to explain its interactions, while also being optimistic about what reward it can achieve. We define this optimism as a logical constraint between a program and a planner. We study our agent on gridworlds, and on task planning, finding our approach is more sample-efficient compared to deep RL, more compute-efficient compared to ReAct-style agents, and that it can transfer its knowledge across environments by editing its code.

## 1 Introduction

Consider yourself learning to use a new device or play a new game. Given the right prior knowledge, together with relatively few interactions, most people can acquire basic knowledge of how many devices or games work. This knowledge can help achieve novel goals, can be transferred to similar devices or games, and can be communicated symbolically to other humans. How could an AI system similarly acquire, transfer, and communicate its knowledge of how things work? We cast this as learning a world model: a mapping, called the *transition function*, that predicts the next state of affairs, given a current state and action [63, 55, 26]. Our proposed solution is an architecture that synthesizes a Python program to model its past experiences with the world, effectively learning the transition function, and which takes actions by planning using that world model.

In theory, world models have many advantages: they allow reasoning in radically new situations by spending more time planning different actions, and accomplishing novel goals by just changing the reward function. Representing world knowledge as code, and generating it from LLMs, brings other advantages. It allows prior world knowledge embedded in the LLM to inform code generation, allows sample-efficient transfer across tasks by reusing pieces of old programs, and allows auditing the system's knowledge, because programming languages are designed to be human-interpretable.

But there are also steep engineering challenges. Learning the world model now requires a combinatorial search over programs to find a transition function that explains the agent's past experiences. Obtaining those experiences in the first place requires efficient exploration, which is difficult in long horizon tasks with sparse reward. To address the challenge of efficient exploration, we introduce a new learning objective that prefers world models which a planner thinks lead to rewarding states, particularly when the agent is uncertain as to where the rewards are. To address the program search problem, we show how curriculum learning can allow transfer of knowledge across environments, making combinatorial program synthesis more tractable by reusing successful pieces of old programs.

Fig. 1 diagrams the resulting architecture, which we cast as model-based reinforcement learning (MB RL). In Fig. 2 we position this work relative to deep RL as well as LLM agents. In contrast to deep

38th Conference on Neural Information Processing Systems (NeurIPS 2024).

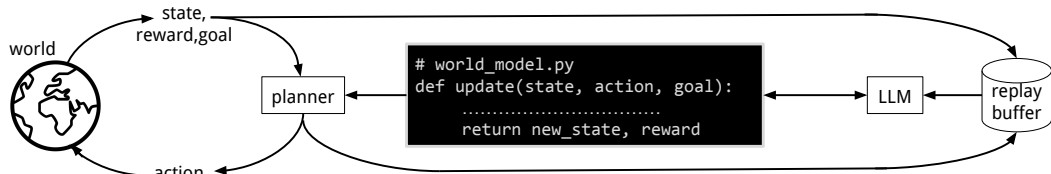

Figure 1: Overall agent architecture. The agent also inputs a goal in natural language

| | Priors | Sample complexity | World model representation | Inputs | LLM calls/task |
|---|---|---|---|---|---|
| Deep MB RL | Low | High | Neural, learned | High-dim | – |
| LLM Agents | High | Zero* | Neural, fixed | Symbolic | $\mathcal{O}(T)$ |
| Ours | High | Low | Symbolic, learned | Symbolic | $\mathcal{O}(1)$ |

Figure 2: Qualitative comparison of our method against deep model-based RL and LLM agents (ReAct, RAP, etc: Yao et al. [77], Hao et al. [29], Zhao et al. [78], Liu et al. [41]). Sample complexity refers to the number of environment interactions needed to learn a world model (*LLM agents do not update their world model). LLM calls/task is the number of LLM calls needed to solve a new task in a fixed environment, amortized over many such tasks, as a function of the maximum episode length $T$. Asymptotically, after learning a world model, our method can accomplish new tasks by only at most one LLM query to update the reward function.

RL [55, i.a.], we view the world model as something that should be rapidly learnable and transferable. Central to our work is a particular claim about how an LLM should relate to world models. In our setup, the LLM does not simulate the world, but instead *builds* a simulation of the world. This should be contrasted with LLM agents such as ReAct [77] where the LLM plays the role of a world model by reasoning about different actions and their consequences. We also do not expect the LLM to perform planning, which they are known to struggle with [68]. Instead, we require the LLM to possess fuzzy prior knowledge of how the world *might* be, together with the programming skill to code and debug a transition function. Our use of LLMs is closest to [24, 74, 79], which generate planning productions, which can be seen as a particular kind of world model. Overall though, our problem statement is closer to [13, 17, 67], which learn world models in domain-specific programming languages. We further show how to use Turing-complete languages like Python—which we believe important for general-purpose learners—and we also study efficient transfer learning and exploration strategies.

We make the following three contributions:

1. An architecture, which we call WorldCoder, for learning world models as code. The architecture supports learning that is more transferable, interpretable, and dramatically more sample-efficient compared to deep RL, and also more compute-efficient than prior LLM agents.

2. A new learning objective for program-structured world models that favors *optimism in the face of (world model) uncertainty* (Section 2.2). We show theoretically and empirically that this learning objective generates goal-driven exploratory behavior, which can reduce by orders of magnitude the number of environment interactions needed to obtain reward

3. An analysis of different forms of transfer of world models, finding that code can be quickly adapted to new world dynamics within grid-world and robot task planning domains.

## 2 Methods

### 2.1 Problem statement and core representations

We start with the standard MDP formalism but modify it in three important ways. First, we assume a goal is given in natural language. The goal could be something specific such as "pickup the ball", or underspecified, such as "maximize reward." Second, we restrict ourselves to deterministic environments, and assume that environment dynamics are fixed across goals. Third, we assume an episodic MDP with the current episode terminating upon reaching the goal.

We formalize this as a Contextual Markov Decision Process (CMDP: Hallak et al. [28]), which is a tuple $(C, S, A, M)$ where $C$ is a set of contexts (i.e. goals), $S$ is a set of states, $A$ is a set of actions, and $M$ is a function mapping a context $c \in C$ to a Markov Decision Process (MDP). The context-conditioned MDP, $M(c)$, is a tuple $(S, A, T, R^c, \gamma)$ with transition function $T : S \times A \to S$, discount factor $\gamma$, and reward function $R^c : S \times A \times S \to \mathbb{R}$. The transition function does not depend on the context. The objective is to select actions to maximize cumulative discounted future reward, which is given by $\sum_{t=0}^{\infty} \gamma^t r_t$, where $r_t$ is the reward received $t$ timesteps into the future. Termination is modeled by assuming there is a special absorbing state.

**State representation.** Motivated by robot task planning [20, 59, 57] we represent states as sets of objects, each with a string-valued field `name`, fields `x` and `y` for its position, and additional fields depending on the object type. For example, if `name="door"`, then there are two Boolean fields for if the door is open/closed and locked/unlocked. This can be seen as an Object Oriented MDP [15].

**Representing world models as code.** The agent uses Python code to model the transition and reward functions. Mathematically we think of this Python code as a tuple $(\hat{T}, \hat{R})$ of a transition function $\hat{T} : S \times A \to S$ and a reward model $\hat{R} : C \to (S \times A \times S \to \mathbb{R} \times \{0, 1\})$. Note again that the reward depends on the context, and returns an extra Boolean indicating whether the goal has been reached, in which case the current episode terminates. Both functions are implemented as separate Python subroutines, which encourages disentangling the dynamics from the reward.

## 2.2 The world model learning problem

What objectives and constraints should the world model satisfy? Clearly, the learned world model should explain the observed data by correctly predicting the observed state transitions and rewards. This is a standard training objective within model-based RL [62].

One less obvious learning objective is that *the world model should suffice to plan to the goal.* Given two world models, both consistent with the observed data, the agent should prefer a model which implies it is possible to get positive reward, effectively being optimistic in the face of uncertainty about world dynamics. Assuming the low-data regime, there will be multiple world models consistent with the data: preferring an optimistic one guarantees that the agent can at least start making progress toward the goal, even if it later has to update its beliefs because they turned out to be too optimistic.

Concretely, the agent collects a dataset $\mathcal{D}$ of past environment interactions, each formatted as a tuple $(s, a, r, s', c, d)$ of current state $s$, next state $s'$, action $a$, and reward $r$ in context $c$ with $d$ indicating if the episode ended upon reaching $s'$. (The variable $d$ should be read as "done.") The agent also stores the initial state $s_0$ and context $c$ of each episode, so that it can prefer world models that can reach the goal from an initial state. The learning problem is to construct Python programs implementing a transition function $\hat{T}$ and reward model $\hat{R}$ satisfying constraints $\phi_1$ and $\phi_2$, defined below:

$$\textit{fit data} \qquad \phi_1\left(\mathcal{D}, \hat{T}, \hat{R}\right) = \quad \forall (s, a, r, s', c, d) \in \mathcal{D} : (\hat{T}, \hat{R}) \vdash (s, a, r, s', c, d) \qquad (1)$$

$$\text{where } (\hat{T}, \hat{R}) \vdash (s, a, r, s', c, d) \text{ if } \hat{T}(s, a) = s' \wedge \hat{R}(c)(s, a, s') = (r, d)$$

$$\textit{optimism} \qquad \phi_2\left(s_0, c, \hat{T}, \hat{R}\right) = \quad \exists a_1, s_1, a_2, s_2, ..., a_\ell, s_\ell \qquad (2)$$

$$\forall i \in [\ell] : \hat{T}(s_{i-1}, a_i) = s_i \wedge \exists r > 0 : \hat{R}(c)(s_{\ell-1}, a_\ell, s_\ell) = (r, 1)$$

where $s_0, c$ is the initial state/context of the episode, and the turnstile ($\vdash$) should be read as meaning that a given program entails a given logical constraint or predicts a given replayed experience.

Constructing a world model satisfying $\phi_1 \wedge \phi_2$ is a program synthesis problem. We solve it by prompting an LLM with a random subset of $\mathcal{D}$ and asking it to propose a candidate program. If the resulting program does not fit all the data, the LLM is backprompted with members of $\mathcal{D}$ inconsistent with its latest program, and asked to debug its code, following [7, 47]. Sec. 2.5 details this process.

## 2.3 Understanding optimism under uncertainty

Optimism under uncertainty is a pervasive principle throughout learned decision-making [63], including model-based RL [75, 40]. We contribute a new instantiation of that principle as a logical constraint between a program and a planner, and which we show can be made compatible with

LLM-guided program generation. Below we give formal and intuitive guides to how this logical constraint plays out in our context.

**Exploration guided by goal-driven behavior.** Before ever receiving reward, $\phi_2$ forces the agent to invent a reward function. This reward function will be something that the agent believes it could achieve, but has not already achieved: roughly, something within its 'zone of proximal development' [71]. After probing its zone of proximal development, the agent updates its model based on the experience of actually trying to achieve a new goal (these model updates come from enforcing $\phi_1$). This exploration continues until finishing an episode with positive reward.

For example, suppose the context (goal) is "open the door with the key", and the agent has not yet achieved any reward, nor has it ever even found the key. Then $\phi_2$ would (for example) prefer $\hat{R}$'s which reward touching the key. Even if touching the key does not *actually* give reward, the agent can make good progress by pretending it does.

**Formal analysis of exploratory behavior.** We prove that a good-enough world model is guaranteed to be discovered in time that is polynomial in several key quantities, defined below. In contrast to prior work on model-based optimism under uncertainty [75, 3, 60, 31, 40], our analysis is not probabilistic, and instead leverages combinatorial and logical properties of the solution space, because our models are represented as discrete programs.

**Definition 2.1.** A dataset $\mathcal{D}$ is **mutually independent** w.r.t. a solution space $\mathcal{M}$, denoted, $\mathcal{D} \perp\!\!\!\perp \mathcal{M}$, iff for all data points in $d \in \mathcal{D}$, there exists a solution in $\mathcal{M}$ which explains all the data *except for* $d$:

$$\forall d \in \mathcal{D}, \exists m \in \mathcal{M}, m \nvdash d \wedge (\forall d' \in \mathcal{D} - \{d\}, m \vdash d').$$

**Definition 2.2.** The **logical dimensionality** of a model space $\mathcal{M}$, written $K_{\perp\!\!\!\perp \mathcal{M}}$, is the size of the biggest dataset that is mutually independent of $\mathcal{M}$:

$$K_{\perp\!\!\!\perp \mathcal{M}} = \max\{|\mathcal{D}| : \forall \mathcal{D} \text{ where } \mathcal{D} \perp\!\!\!\perp \mathcal{M}\}$$

Within this paper the model class corresponds to the cross product of possible transition functions and possible reward functions, denoted $\mathcal{T}$ and $\mathcal{R}$, respectively, so $\mathcal{M} = \mathcal{T} \times \mathcal{R}$.

**Definition 2.3.** The **diameter** of a deterministic episodic MDP, written $D_{S,A,T}$, is the maximum shortest path length between all pairs of states:

$$D_{S,A,T} = \max_{s,s' \in S} \min_{p \text{ a path from } s \text{ to } s'} \text{len}(p)$$

Putting these together, Appendix A shows that we can learn in time polynomial w.r.t. the diameter (a property of the true MDP) and the logical dimensionality (a property of the model space):

**Theorem 2.4.** *Assume an episodic MDP* $(S, A, T, R, \gamma)$. *Assume an agent acting according to an optimal planner operating over world model* $(\hat{T}, \hat{R}) \in \mathcal{T} \times \mathcal{R}$ *satisfying* $\phi_1 \wedge \phi_2$, *and that the true MDP is in the agent's model class:* $T \in \mathcal{T}$ *and* $R \in \mathcal{R}$. *Then the maximum number of actions needed to achieve the goal (exit with positive reward) is* $D_{S,A,T} \times (K_{\mathcal{T} \times \mathcal{R}} + 1)$.

Intuitively, the polynomial sample complexity is possible because the agent systematically finds independent data points, which serve as counter-examples to the agent's current model. At most $K_{\perp\!\!\!\perp \mathcal{T} \times \mathcal{R}}$ such counterexamples are needed, each requiring at most $D_{S,A,T}$ actions.

**Following natural-language instructions.** After learning how the world works, our agent should be able to receive natural language instructions and begin following them immediately without further learning from environmental interactions. Mathematically, given a learned program $\hat{R}$ that implements reward functions for previously seen goals, together with a new goal $c$ where $c \notin \text{domain}(\hat{R})$, the agent should update its reward model in order to cover $c$.

This zero-shot generalization to new goals occurs as a consequence of enforcing optimism under uncertainty ($\phi_2$ in Eq. 2). Upon observing a new context $c$, the constraint $\phi_2$ is violated. This triggers debugging $\hat{R}$ so that it covers $c$, subject to the constraint that $\hat{R}(c)$ allows reaching a goal state.

Given the importance of instruction-following abilities, we use a different prompt for trying to enforce $\phi_2$ upon encountering a new goal (Appendix Sec. F.5). This prompt is based on retrieving previously learned reward functions. This retrieval strategy assumes similar goals have similar reward structures, so that the LLM can generalize from old reward functions in $\hat{R}$ when generating the new $\hat{R}(c)$. If the LLM makes a mistake in predicting the reward function, then the agent can recover by subsequent rounds of program synthesis, which update $\hat{R}(c)$ based on interactions with the environment.

**Algorithm 1** WorldCoder Agent Architecture

---

**Hyperparam:** $\epsilon$, random exploration probability (default to $5\%$)
**Hyperparam:** MINDATASIZE, min # actions before learning begins (default to 10)

$\quad \mathcal{D}, \mathcal{D}_{sc} \leftarrow \varnothing, \varnothing$ $\qquad\qquad\qquad$ ▷ replay buffer. $\mathcal{D}_{sc}$ holds initial states and contexts needed for $\phi_2$
$\quad \hat{T}, \hat{R} \leftarrow \texttt{null}, \texttt{null}$ $\qquad\qquad\qquad\qquad\qquad\qquad$ ▷ init empty world model
$\quad$ **loop forever through episodes:**
$\qquad c \leftarrow$ EPISODEGOAL() $\qquad\qquad\qquad\qquad\qquad\qquad$ ▷ get context (goal)
$\qquad s_0 \leftarrow$ CURRENTSTATE() $\qquad\qquad\qquad\qquad\qquad\quad$ ▷ record initial state
$\qquad \mathcal{D}_{sc} \leftarrow \mathcal{D}_{sc} \cup \{(s_0, c)\}$ $\qquad\qquad\qquad$ ▷ Replay buffer of initial conditions for $\phi_2$
$\qquad$ **loop until episode ends:**
$\qquad\quad s \leftarrow$ CURRENTSTATE()
$\qquad\quad$ **if not** $\phi_1 \wedge \phi_2$ **and** $|\mathcal{D}| \geq$ MINDATASIZE **then**
$\qquad\qquad \hat{T}, \hat{R} \leftarrow$ SYNTHESIZE($\hat{T}, \hat{R}, \mathcal{D}, \mathcal{D}_{sc}$) $\qquad\qquad\qquad$ ▷ Section 2.5
$\qquad\quad$ **with probability** $\epsilon$ **do**
$\qquad\qquad a \leftarrow$ RANDOMACTION() $\qquad\qquad\qquad\qquad$ ▷ $\epsilon$-greedy explore
$\qquad\quad$ **else**
$\qquad\qquad a \leftarrow$ PLAN($s, \hat{T}, \hat{R}(c)$) $\qquad\qquad\qquad\qquad$ ▷ Value Iteration
$\qquad\quad s', r, d \leftarrow$ ENV.STEP($a$) $\qquad\qquad\qquad\qquad$ ▷ take action in state $s$
$\qquad\quad \mathcal{D} \leftarrow \mathcal{D} \cup \{(s, a, r, s', c, d)\}$ $\qquad\qquad\qquad\qquad$ ▷ record experience

---

## 2.4 Overall architecture

Ultimately our world models exist to serve downstream decision-making: and in turn, taking actions serves to provide data for learning better world models. There are many architectures for combining acting with world-model learning, such as via planning [55], training policies in simulation [26], or hybrid approaches [62]. We use the architecture shown in Algorithm 1. At a high level, it initially performs random exploration to initialize a dataset of environmental interactions; it updates its world model using the program synthesis algorithm of Sec. 2.5; and, past its initial exploration phase, performs planning. Different planners are possible, and we use depth-limited value iteration (in simple domains) and MCTS (for more complex domains).

## 2.5 Program Synthesis via Refinement

One approach to program synthesis is to have an LLM iteratively improve and debug its initial outputs [7, 47], which we call *refinement*. For world models as programs, this means prompting the LLM with an erroneous program it previously generated, together with state-action transitions that program cannot explain, and prompting it to fix its code. This process repeats until the program satisfies $\phi_1 \wedge \phi_2$. Refinement can be very successful when the target program has many corner-cases, each of which can be inferred from a few examples, because the LLM can incrementally grow the program driven by each failed test case. This is exactly the case for world models, which might need to handle a wide range of objects and their interactions, but typically don't demand intricate algorithms. Refinement also allows computationally efficient transfer between environments, because a new world model can be built by refining an old one, instead of programming it from scratch.

We use a concurrently developed algorithm called REx to determine which program to refine next [65]. REx prioritizes refining programs that appear more promising, which in the context of world model learning, we instantiate by defining by a heuristic $h$, which measures progress towards satisfying $\phi_1 \wedge \phi_2$:

$$h(\hat{T}, \hat{R}) = \frac{\sum_{x \in \mathcal{D}} \mathbb{1}\left[\rho \vdash x\right] + \mathbb{1}\left[\phi_1\left(\mathcal{D}, \hat{T}, \hat{R}\right)\right] \times \sum_{s_0, c \in \mathcal{D}_{sc}} \left[\mathbb{1}\left[\phi_2(s_0, c, \hat{T}, \hat{R})\right]\right]}{|\mathcal{D}| + |\mathcal{D}_{sc}|} \tag{3}$$

The above heuristic computes the fraction of the replay buffer which is consistent with $\phi_1$, and the fraction which is consistent with $\phi_2$. (We incentivize satisfying $\phi_1$ first by multiplying the average accuracy on the optimism objective by the indicator $\mathbb{1}[\phi_1(\mathcal{D}, \hat{T}, \hat{R})]$.) REx also prioritizes refining programs that have not been refined very many times, and uses a bandit formulation to balance (1) exploring programs that have not been refined very many times against (2) exploiting by refining programs that are better according to $h$. See [65] for more details.

We implement this refinement process using GPT-4 because recent work [47] finds it is the strongest model for repairing and improving code (as opposed to just generating code from scratch). Using this technique and this LLM we can generate world models with 250+ lines of code.

## 3 Experimental Results

We study our system in three environments, Sokoban, Minigid, and AlfWorld, with the goal of understanding the sample efficiency and computational efficiency of the learner, especially when transferring knowledge across environments, as well as the impact of optimism under uncertainty.

**Sokoban** is a puzzle-solving task where the agent pushes boxes around a 2d world, with the goal of pushing every box onto a target (Fig. 3A). Solving hard Sokoban levels is a challenging planning task that has received recent attention from the planning and RL communities [10, 11, 37, 19, 18, 54]. Unlike these other works, our emphasis is not on solving the hardest Sokoban levels. Instead, we wish to show that our agent can rapidly achieve basic competence. Master-level play could then be achieved via any of the cited works that focus on sophisticated planning and search.

Starting with only the natural-language goal of "win the game", our agent builds a world model over the first 50 actions. The resulting code is human-understandable (Appendix D), and generalizes to solving levels with more boxes (Fig. 3B). While the system cannot solve very hard Sokoban levels (eg, 5+ boxes), that is an artifact of the difficulty of planning, and could be addressed by plugging the world model into any of the techniques cited above. In contrast to this work, both model-based and model-free deep RL require millions of experiences to solve basic levels (Fig. 3D).

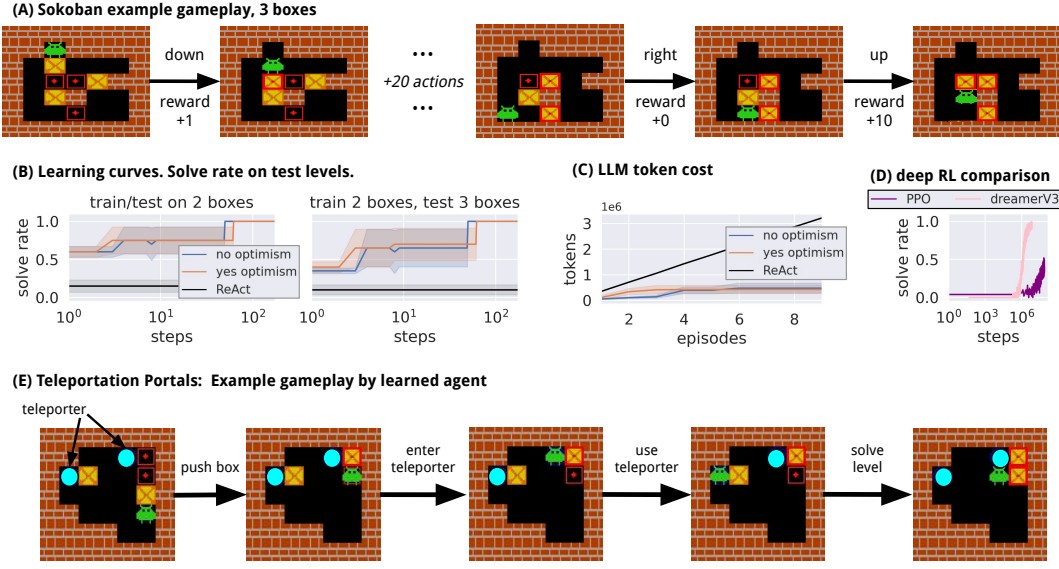

Figure 3: (A) Sokoban domain (per-step reward of -0.1 ellided from figure). (B) Learning curves. ReAct has the same pretrained knowledge of Sokoban but cannot effectively play the game. (C) Our method has different asymptotic LLM cost compared to prior LLM agents, which consume LLM calls/tokens at every action. (D) Deep RL takes >1 million steps to learn 2-box Sokoban. (E) Nonstandard Sokoban with teleport gates

Almost surely, an important reason why our system learns quickly is because the underlying LLM already knows about Sokoban from its pretraining data, and can quickly infer that it is playing a similar game. However, simply knowing about Sokoban does *not* suffice for the LLM to play the game, as demonstrated by the poor performance of ReAct (Fig. 3B). ReAct is a baseline which prompts the LLM with the state-action history, then asks it to think step-by-step (Reason) and before predicting an action (Act). Quantitatively, ReAct succeeds on only $15\% \pm 8\%$ of basic levels, showing that pretrained knowledge of Sokoban does not, by itself, allow strong play. ReAct-style architectures [29, 78, 41, i.a.], also require expensive LLM calls *at every action*, and so asymptotically their cost grows linearly with the number of actions taken. Our approach has different asymptotics:

after front-loading 400k LLM tokens (about $15), it can issue as many actions as needed without subsequent LLM queries (Fig. 3C).

To further demonstrate that pretrained knowledge of Sokoban cannot explain the success of our system, we modify the game to include extra rules by adding a pair of warp gates/teleportation portals to each level which the agent can use to instantly transport itself (Fig. 3E). The teleportation portals have subtle dynamics, because they become deactivated whenever they are blocked by a Sokoban box. Our agent continues to be able to learn in this environment, including modeling the blocking behavior of the teleportation gates, and learns to use the teleporters to more rapidly solve levels.

On Sokoban, the optimism under uncertainty objective ($\phi_2$, orange curves in Fig. 3B) has little effect: Sokoban has a dense reward structure that allows easy learning through random exploration. Next we consider problems with sparse rewards and natural language instructions, which the optimism objective is designed to handle.

**Minigrid.** To better understand the transfer learning and instruction following aspects of our approach, we next study Minigrid [9, 8], a suite of grid games designed for language-conditioned RL. Minigrid environments include objects such as keys, doors, walls, balls, and boxes.

Fig. 4 illustrates results for our agent playing a sequence of minigrid environments, while Appendix A.1 gives a walk-through of an example learning trajectory. The agent interacts with each environment episodically through different randomly-generated levels. We order the environments into a curriculum designed to illustrate different forms of transfer learning. For example, when transferring from the first to the second environment, the agent needs to extend its knowledge to incorporate new objects and their associated dynamics (keys and doors). Learning about these new objects requires extra environment interactions, during which the agent experiments with the objects to update its transition function. In contrast, the third environment presents no new dynamics, but instead introduces new natural-language goals. Our agent can follow these natural-language instructions by enforcing optimism under uncertainty ($\phi_2$).

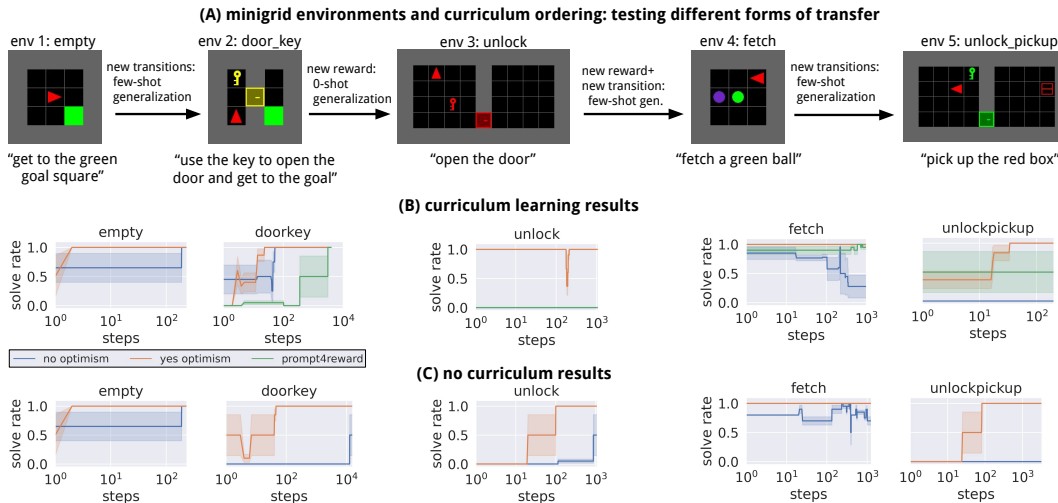

Figure 4: (A) Minigrid environments ordered into a curriculum that tests different kinds of transfer learning. (B) Transfer learning performance, compared with (C) performance when solving each environment independently. Appendix Fig. 6: deep RL comparison.

Transfer is especially helpful in quickly solving new environments (Fig. 4B). Without transfer, more episodes of exploration are needed to collect experience to build a good world model. However, optimism under uncertainty ($\phi_2$) helps in non-transfer setting by promoting goal-directed exploration, and in fact, absent transfer, it is only with $\phi_2$ that WorldCoder can solve the harder environments. Optimism is also necessary for zero-shot generalization to new goals (Fig. 4B, env 3).

To better understand the importance of $\phi_2$—which both encourages exploration and enables natural-language instruction following—we contrast against a version of our approach which simply prompts GPT4 to generate a new reward function upon receiving a new goal (green in Fig. 4B, labelled 'prompt4reward'). Theoretically this suffices to follow new natural-language instructions, provided

the LLM is reliable. Surprisingly, this ablation struggles to follow new instructions (e.g., transfer from env 2→3 in Fig. 4A), showing the importance of $\phi_2$ in correcting mistakes made by the LLM when generating reward functions.

**AlfWorld.** To test the scalability of our method we work with the AlfWorld robot task planning domain. This is a household robot planning environment with a variety of different kinds of objects, such as microwaves, cabinets, utensils, food, and different rooms the robot can navigate. We convert AlfWorld into a MDP by representing the state as a set of fluents (in the style of PDDL [52]), and study our agent as it progresses through representative AlfWorld tasks.

In Fig. 5 we find that our method learns models of picking up and placing objects, and then improves its model by learning how to use cooking appliances such as refrigerators and stoves. This stresses the scalability of the exploration and program synthesis methods: We find that we can synthesize a world model containing 250+ lines of code, which serves as an adequate model of how AlfWorld works (Appendix E), and that the optimism under uncertainty objective is necessary for nonzero performance on all of these tasks. Typically the agent solves the task in the first episode, but requires around 20 exploratory steps to get reward, with subsequent episodes serving to make small corrections to the world model. Fig. 5 indicates with arrows the episodes where these small corrections occur.

These tasks have relatively long horizons, so we use a more sophisticated planner based on MCTS. Because these tasks also have sparse rewards, we engineer a domain-general partial reward function that incentivizes actions and states which have more textual overlap with the natural-language goal, based on established document retrieval metrics (BM25 [50], Appendix B). This shows that our framework can interoperate with different kinds of planners.

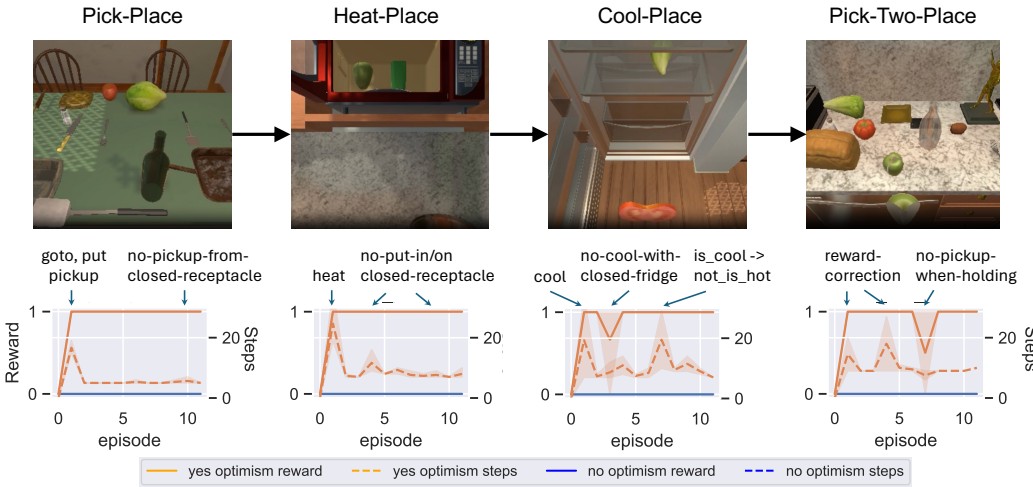

Figure 5: AlfWorld environments and tasks. Each learning curve shows average reward at each episode (solid line) and how many steps the episode took (dashed), averaged over 3 seeds. Curves annotated with arrows and text explaining what was learned at each episode. Optimism objective is necessary for any non-zero performance.

# 4 Related Work

**World models.** World modeling is a classic paradigm for decision-making: It is the basis of model-based RL [27, 26], task-and-motion-planning in robotics [32], and it is corroborated by classic behavioral experiments suggesting biological decision-making features a similar architecture [66]. In practice, world models are hard-won, requiring either large volumes of training data [26, 25], or careful hand-engineering [32] (cf. Mao et al. [44], Konidaris et al. [36]).

**Neurosymbolic world models,** such as Cosmos and NPS [56, 22], learn a factored, object-based neural world model. This factoring helps compositional generalization—like in our work—but importantly they can learn from raw perception, but at the expense of transfer and sample complexity. Combining our work with these others might be able to get the best of both.

**LLMs as a world model.** Whether LLMs can model the world is an open question, but there is evidence that, given the right training data in large quantities, transformers can act as decent world models, at least within certain situations [38, 76, 45]. These works aim to learn a rich but frozen world model from a relatively large volume of examples. We tackle a different problem: building a simple world model on-the-fly from a modest amount of data.

**LLMs for building world models.** Recent works [79, 74, 24] consider using LLMs to generate planning operators: a kind of world model that as abstract, symbolic, and expressed in a domain-specific programming language for planning (cf. DECKARD [46], another LLM system which generates state-machine world models). In these works, the primary driver of world-model generation—what the LLM first inputs—is natural language describing affordances and goals. Our work considers a different problem: building world models from first-and-foremost from interacting with the environment. In practice, agents have knowledge both from language and from acting in the world, and so these families of works should be complementary.

**LLMs for decision-making** is an emerging paradigm that includes ReAct [77] and many others [29, 78, 41, 1, i.a.], which directly use LLMs to issue actions and reason about their consequences in the world. For instance, ReAct works by prompting the LLM to think step-by-step and then predict an action. To the extent that these methods use a world model, it is implicitly encoded within the weights of a neural network. We instead build an explicit world model.

**Programs as Policies.** Instead of learning a world model, one can learn a policy as a program. The first wave of these works [69, 70] considered domain-specific languages, while recent LLM work [73, 39, 61] uses more flexible general-purpose languages like Python. An advantage of learning a policy is that it does not need to model all the details of the world, many of which may be irrelevant to decision making. A disadvantage is that policies cannot readily generalize to new goals—unlike world models, which can be used by a planner to achieve a variety of objectives. Relatedly, other recent work considers synthesizing programs that implement reward functions [42], and then generating a policy with conventional deep RL.

**Programs as world models.** We are strongly inspired by existing program synthesis algorithms for constructing world models from state-action trajectories [13, 17, 67]. We believe that this family of methods will not be generally applicable until they can support general-purpose Turing-complete programming languages: So far these works have used restricted domain-specific languages, but we show that a general-purpose computational language, like Python, can be used to learn world models, which we hope expands the scope of this paradigm. We also show how to bias learning toward goal-directed behaviors, and how to support transfer across environments and goals. Last, we simplify the core program synthesis algorithm: the cited prior works required relatively intricate synthesis algorithms, which we can avoid by using LLMs as general-purpose synthesizers. We hope our work can help make this paradigm simpler and more general.

Other works have also explored how humans can manually provide knowledge to RL agents via source code: e.g., RLLang [51] uses programs to specify parts of policies and world models, which could be combined with our system to integrate prior knowledge.

**Exploration & Optimism in the face of (model) uncertainty.** Being optimistic about actions with uncertain consequences is common in decision-making, including in model-based RL [63, 40, 75, 3, 60, 31]. We mathematically instantiate that principle in a new way—as a logical constraint between a program and a planner—and show how it can work with LLM-guided discrete program search.

## 5 Limitations and Open Directions

Our work has important limitations, and naturally suggests next steps. Currently we assume deterministic dynamics, which could be addressed by synthesizing probabilistic programs [14, 21]. Given recent advances in synthesizing probabilistic programs [53], together with advances in using LLMs for deterministic code, this limitation seems nontrivial but surmountable.

By representing knowledge as code, our approach delivers better sample efficiency and transferability, but at high cost: Our states must be symbolic and discrete, whereas the real world is messy and continuous. While the obvious response is that the agent can be equipped with pretrained object detectors—a common assumption in robotic task planning [36, i.a.]—alternative routes include

multimodal models [30] and using neurosymbolic programs [6, 56, 64] to bridge the gap between perception and symbol processing, which might be more robust to missing or misspecified symbols.

Last, our method uses only a very basic mechanism for growing and transferring its knowledge. Instead of prompting to debug its code, we could have built a library of reusable subroutines and classes shared across different environments and goals, reminiscent of library learning systems [16, 72, 23, 4], which refactor their code to expose sharable components. Further developing that and other ways of managing and growing symbolic knowledge about the world remains a prime target for future work.

**Acknowledgements.** This work was supported by NSF grant #2310350 and a gift from Cisco.

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

# A Theoretical analysis of Sample Efficiency using optimism under uncertainty ($\phi_2$)

The optimism under uncertainty objective ($\phi_1 \wedge \phi_2$ in Sec. 2) is much more sample efficient than the traditional data-consistency objective ($\phi_1$) as shown in Figure 4 in our experiments. We will later show a learning trajectory of it in the MiniGrid-UnlockPickup-v0 environment in Appendix A.1. This UnlockPickup environment is difficult for exploration due to its large search space. PPO failed to gain enough positive-reward supervision signals after $3 \times 10^8$ actions as shown in Figure 6. Our method failed *without* the optimism objective as well. Nevertheless, our method *with* the optimism objective learned the correct world model from scratch with no more than 100 actions due to its better sample efficiency in exploration.

Here we provide a simple theorem stating that significantly better sample efficiency is theoretically guaranteed when using this objective (polynomial w.r.t the diameter of the state space and the solution space) following an intuitive observation:

**Observation.** When planning with a world model that satisfies the $\phi_1 \wedge \phi_2$ there are only two possible outcomes:

- Either the model is correct: the agent achieves the goal successfully;
- Or the model is incorrect: the agent then must find a counter-example to its world model's prediction and gain more knowledge of the environment.

It shows that the world model that satisfies $\phi_1 \wedge \phi_2$ is either correct or is efficient in guiding the agent to find a counter-example of the current world model. Counter-example are precious because, when in search of the correct world model, each interaction data $(s, a, s', r, d)$ in the replay buffer implies a constraint to satisfy for the potentially correct world models in the space of world models. Collecting data that can be explained by all potentially correct world models is useless because it implies no stricter constraints than the current ones. Only through the counter-examples which cannot be explained by the current world model, the size of the set of the potentially correct world models can become smaller and smaller, which eventually leads to a set that only contains the good-enough world models that guide the agent towards the goal.

The optimism under uncertainty objective is much more sample efficient to get those valuable counter-examples than random exploration. For deterministic environments, the number of actions to find a counter-example with a world model that satisfies $\phi_1 \wedge \phi_2$ is guaranteed to be smaller than the diameter of the state space, $D_S$, when equipped with an optimal planner (by definition). Random exploration needs exponentially more number of actions to find this counter-example in the worst case. (Note that we *do not* assume free reset to all possible states, so the agent needs to go through the trajectories from the initial state to each target state.)

More formally, we show that the maximum number of actions to learn/find a good-enough world model is polynomial w.r.t. intrinsic properties of the state space and model space:

**Definition A.1.** A dataset $\mathcal{D}$ is **mutually independent** w.r.t. a solution space $\mathcal{M}$, denoted, $\mathcal{D} \perp\!\!\!\perp \mathcal{M}$, if and only if for all data points in $d \in \mathcal{D}$, there exists a solution in $\mathcal{M}$ which explains all the data *except for* $d$:
$$\forall d \in \mathcal{D}, \exists m \in \mathcal{M}, m \nvdash d \wedge (\forall d' \in \mathcal{D} - \{d\}, m \vdash d').$$
**Definition A.2.** The **logical dimensionality of** $\mathcal{M}$, denoted as $K_{\perp\!\!\!\perp \mathcal{M}}$, is
$$K_{\perp\!\!\!\perp \mathcal{M}} = \max\{|\mathcal{D}| : \forall \mathcal{D} \text{ where } \mathcal{D} \perp\!\!\!\perp \mathcal{M}\}$$

It is straightforward to prove that this is at most $|\mathcal{M}|$:

**Lemma A.3.** $K_{\perp\!\!\!\perp \mathcal{M}} \leq |\mathcal{M}|$, *i.e., the maximum size of the mutually independent data sets w.r.t. to a solution space is smaller than the size of the solution space.*

Proof: each data point $d \in \mathcal{D}$ at least exclude one more unique solution in the solution space than the other data points. Otherwise, the data set is not mutually independent w.r.t. the solution space.

Note that this is the loosest upper bound of $K_{\perp\!\!\!\perp \mathcal{M}}$. We assume nothing about the model space and the learning algorithm. In practice, $K_{\perp\!\!\!\perp \mathcal{M}}$ should be much smaller than the size of the solution space. For example, for linear models, we only need $n$ independent data points to characterize the correct solution for $d \in \mathbb{R}^n$.

**Theorem A.4.** *Assume an episodic MDP* $(S, A, T, R, \gamma)$. *Assume an agent acting according to an optimal planner operating over world model* $(\hat{T}, \hat{R}) \in \mathcal{T} \times \mathcal{R}$ *satisfying* $\phi_1 \wedge \phi_2$, *and that the true MDP is in the agent's model class:* $T \in \mathcal{T}$ *and* $R \in \mathcal{R}$. *Then the maximum number of actions needed to achieve the goal (exit with positive reward) is* $D_{S,A,T} \times (K_{\mathcal{T} \times \mathcal{R}} + 1)$.

Proof Outline:

*Lemma $\alpha$:* Each replay buffer data set, $\mathcal{D}$, can be represented by its mutually independent data subset, $\mathcal{D}_{\perp\!\!\!\perp \mathcal{T} \times \mathcal{R}}$, from the perspective of finding the correct world model, i.e.,

$$\mathcal{D}_{\perp\!\!\!\perp \mathcal{T} \times \mathcal{R}} \subseteq \mathcal{D} \bigwedge D_{\perp\!\!\!\perp \mathcal{T} \times \mathcal{R}} \perp\!\!\!\perp \mathcal{T} \times \mathcal{R} \bigwedge \forall (\hat{T}, \hat{R}) \in \mathcal{T} \times \mathcal{R}, (\hat{T}, \hat{R}) \vdash \mathcal{D} \Leftrightarrow (\hat{T}, \hat{R}) \vdash \mathcal{D}_{\perp\!\!\!\perp \mathcal{T} \times \mathcal{R}}.$$

*Lemma $\beta$:* For the replay buffer, $\mathcal{D}^{(t)}$, at any step $t$, if a world model $(\hat{T}^{(t)}, \hat{R}^{(t)})$ satisfies the optimism under uncertainty objective as well as the traditional data-consistency objective, i.e., $(\hat{T}, \hat{R}) \vdash \mathcal{D}^{(t)} \wedge (\hat{T}, \hat{R}) \vdash \phi_2$, then either the word model is correct, which means it can guide the agent to the goal successfully, or the agent finds a counter-example, $d'$, to the current world model in $D_S$ steps when given an optimal planner.

*Lemma $\gamma$:* This counter example, $d'$, is mutually independent to the replay buffer, $\mathcal{D}^{(t)}$, because there is a model $(\hat{T}^{(t)}, \hat{R}^{(t)})$ such that $(\hat{T}^{(t)}, \hat{R}^{(t)}) \nvdash d' \wedge (\hat{T}^{(t)}, \hat{R}^{(t)}) \vdash \mathcal{D}^{(t)}$.

Given these lemmas, we have

$$|D^{(t+D_S)}_{\perp\!\!\!\perp \mathcal{T} \times \mathcal{R}}| \geq |D^{(t)}_{\perp\!\!\!\perp \mathcal{T} \times \mathcal{R}}| + 1$$

and therefore

$$|D^{D_S \times (K_{\perp\!\!\!\perp \mathcal{T} \times \mathcal{R}}+1)}_{\perp\!\!\!\perp \mathcal{T} \times \mathcal{R}}| \geq K_{\perp\!\!\!\perp \mathcal{T} \times \mathcal{R}} + 1 + D^{(0)} \geq K_{\perp\!\!\!\perp \mathcal{T} \times \mathcal{R}} + 1 > K_{\perp\!\!\!\perp \mathcal{T} \times \mathcal{R}}.$$

Assume the world model after $D_S \times (K_{\perp\!\!\!\perp \mathcal{T} \times \mathcal{R}} + 1)$ steps is still incorrect, we then build a mutually independent data set, $D^{(D_S \times (K_{\perp\!\!\!\perp \mathcal{T} \times \mathcal{R}}+1))}_{\perp\!\!\!\perp \mathcal{T} \times \mathcal{R}}$, with size larger than $K_{\perp\!\!\!\perp \mathcal{T} \times \mathcal{R}}$, which is contradictory to its definition. $\qquad\square$

The proofs of Lemma $\alpha$ and $\gamma$ are straightforward by definitions. For example, we can build $\mathcal{D}^{(0)}_{\perp\!\!\!\perp \mathcal{T} \times \mathcal{R}}$ by dropping data points in $\mathcal{D}^{(0)}$ that are not mutually independent to the left subset one by one. We can build $D^{(t+1)}_{\perp\!\!\!\perp \mathcal{T} \times \mathcal{R}}$ by adding the counter-example $d'$ to $D^{(t)}_{\perp\!\!\!\perp \mathcal{T} \times \mathcal{R}}$.

We prove Lemma $\beta$ by construction. Given the definition of the optimism under uncertainty objective, $\phi_2$ in Sec. 2:

$$\phi_2\left(s_0, c, \hat{T}^{(t)}, \hat{R}^{(t)}\right) =$$
$$\exists a_1, s_1, a_2, s_2, ..., a_\ell, s_\ell \ :$$
$$\forall i \in [\ell] : \hat{T}^{(t)}(s_{i-1}, a_i) = s_i \wedge$$
$$\exists r > 0 : \hat{R}^{(t)}(c)(s_{\ell-1}, a_\ell, s_\ell) = (r, 1),$$

there exists a trajectory $a_1, s_1, a_2, s_2, ..., a_\ell, s_\ell$ such that either the model correctly leads the agent to the goal, i.e., $\exists r > 0 : R(c)(s_{\ell-1}, a_\ell, s_\ell) = (r, 1)$, or there exists a counter-example, i.e., $\exists i \in [\ell] : \hat{T}^{(t)}(s_{i-1}, a_i) \neq T(s_{i-1}, a_i) \vee \hat{R}^{(t)}(c)(s_{i-1}, a_i, s_i) \neq R(c)(s_{i-1}, a_i, s_i)$, which means $(\hat{T}^{(t)}, \hat{R}^{(t)}) \nvdash (s_{i-1}, a_i, s_i, r_i, d_i)$. We also have $(\hat{T}^{(t)}, \hat{R}^{(t)}) \vdash D^{(t)}$ due to $(\hat{T}^{(t)}, \hat{R}^{(t)}) \vdash \phi_1$. We thus prove Lemma $\beta$.

## A.1 Example learning trajectory using ($\phi_1 \wedge \phi_2$) MiniGridUnlockPickup

To demonstrate the effectiveness of the optimism objective ($\phi_2$ in Sec. 2) in improving the sample efficiency through guided exploration, we show here an example of the learning trajectories in the MiniGrid-UnlockPickup-v0 environment. In comparison with agents that only use the traditional data-consistency objective ($\phi_1$), which merely relies on random exploration to gain new knowledge of the world through new data, agents that use the extra optimism under uncertainty objective can

- imagine the functionality of actions and their interactions with the necessary tools, in order to achieve the goal, without any real interactions with the environment;

- explore much more efficiently guided by the imagined world model;

- and correct the imagined world model given the newly collected data after the efficient exploration.

The learning trajectory is as follows:

| Original reward function | Original transit function |
|---|---|
| NULL | NULL |

**Random exploration**
No feasible move yet except for rotation.

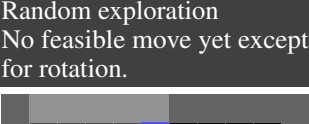
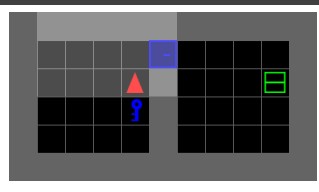

**Data-consistent transit-func**
(figured out how to rotate)

```python
def turn_right(direction):
    if direction == (-1, 0):
        return (0, -1)
    elif direction == (1, 0):
        ...
def turn_left(direction):
    if direction == (-1, 0):
        return (0, 1)
    elif direction == (1, 0):
        ...
def transition(state, action):
    agents = get_entities_by_name(state, "Agent")
    assert len(agents) == 1, "There should be
    exactly one agent in the state"
    agent = agents[0]
    if action == "turn right":
        agent.direction = turn_right(agent.direction
        )
    elif action == "turn left":  # Add a condition
    for the "turn left" action
        agent.direction = turn_left(agent.direction)
    # The other actions does not seem to affect the
    state
    return state
```

**Data-consistent reward-func**
**(random guess given the textual mission)**

```python
def reward_func(state, action, next_state):
    # If the state does not change after action, the
      reward is 0.0 and not done
    if state == next_state:
        return 0.0, False
    agent_n, agent_o = get_entities_by_name(
    next_state, "Agent")[0], get_entities_by_name(
    state, "Agent")[0]
    box_n, box_o = get_entities_by_name(next_state,
    "Box")[0], get_entities_by_name(state, "Box")[0]
    # If the agent's position gets closer to the box
    , give a bigger reward
    if abs(agent_n.x - box_n.x) + abs(agent_n.y -
    box_n.y) < abs(agent_o.x - box_o.x) + abs(
    agent_o.y - box_o.y):
        return 1.0, False
    # If the agent changes direction but not
    position, the reward is 0.0 and not done
    elif agent_o.x == agent_n.x and agent_o.y ==
    agent_n.y and agent_o.direction != agent_n.
    direction:
        return 0.0, False
    # If the agent picks up the green box, give a
    big reward and finish
    elif agent_n.carrying and agent_n.carrying.color
     == "green":
        return 100.0, True
    # If the agent moves but does not make
    significant progress, give small penalty to make
     it efficient
    else:
        return -0.01, False
```

**Goal-driven transit-func**
**(imagine how to move forward without interactions)**
**(imperfect though as not considering the locked door)**

```python
def transition(state, action):
    ...
    elif action == 'move forward':
        dx, dy = agent.direction
        new_x, new_y = X + dx, Y + dy
        if not get_entities_by_name(
        get_entities_by_position(next_state, new_x,
        new_y), 'Wall'):
            agent.x, agent.y = new_x, new_y
    ...
```

**Goal-driven reward-func**
(more efficient representation <- already correct)

```python
def reward_func(state, action, next_state):
    # Get all agent entities in the next state
    next_agents = get_entities_by_name(next_state, '
    Agent')
    # Check if any agent is carrying a Box
    for next_agent_carrying in [agent.carrying for
    agent in next_agents]:
        if isinstance(next_agent_carrying, Box) and
        next_agent_carrying.color == 'green':
            return 1.0, True
    return 0.0, False
```

**Guided by imagined transit-func**
Trying to cross a locked door

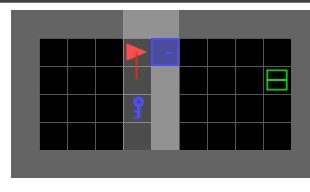

**Data-consistent transit-func**
(figured out the agent cannot step on the wall and the locked door,
but still do not know how to go to the goal.)

```python
ddef transition(state, action):
    ...
    elif action == 'move forward':
        dx, dy = agent.direction
        new_x, new_y = X + dx, Y + dy
        door_at_new_loc = get_entities_by_name(
        get_entities_by_position(next_state, new_x,
        new_y), 'Door')
        if door_at_new_loc:
            door_at_new_loc = door_at_new_loc[0]
            if door_at_new_loc.state == "locked":
                return next_state
        if not get_entities_by_name(
        get_entities_by_position(next_state, new_x,
        new_y), 'Wall'):
            agent.x, agent.y = new_x, new_y
    ...
```

```python
def transition(state, action):
    ...
    elif action == 'pick up':
        items_at_agent_location =
        get_entities_by_position(next_state,
        agent.x, agent.y)
        pickable_items = [item for item in
        items_at_agent_location if item.name not in
        ['Door','Wall', 'Agent']]
        if pickable_items:
            agent.carrying = pickable_items[0]
            next_state.remove(pickable_items[0])
    elif action == 'drop':
        if agent.carrying:
            agent.carrying.x, agent.carrying.y =
            agent.x, agent.y
            next_state.append(agent.carrying)
            agent.carrying = None
    elif action == 'toggle':
        if isinstance(agent.carrying, Key):
            door_at_right = [door for door in
            get_entities_by_position(next_state,
            right_x, right_y) if door.name == 'Door'
            ]
            if door_at_right and door_at_right[0].
            color == agent.carrying.color:
                door_at_right[0].state = 'unlocked'
    return next_state
```

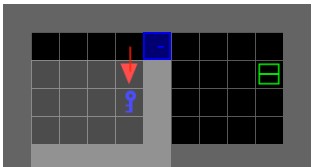

```python
...
def transition(state, action):
    ...
    elif action == 'move forward':
        entities_at_front = [entity for entity in
        get_entities_by_position(next_state, front_x
        , front_y) if entity.name != 'Agent']
        if entities_at_front:
            if entities_at_front[0].name == "Door"
            and entities_at_front[0].state == "
            unlocked":
                agent.x, agent.y = front_x, front_y
            # other entities should prevent agent
            from moving forward
        else:
            agent.x, agent.y = front_x, front_y
    elif action == 'pick up':
        items_at_agent_location =
        get_entities_by_position(next_state,
        agent.x, agent.y)
        pickable_items = [item for item in
        items_at_agent_location if item.name not in
        ['Door','Wall', 'Agent']]
        if pickable_items:
            agent.carrying = pickable_items[0]
            next_state.remove(pickable_items[0])
    ...
```

Goal-driven transit-func
(imagine the agent can also pickup the object in front of it, which is correct, without interactions.

```python
def transition(state, action):
    ...
    elif action == 'pick up':
        items_at_agent_location =
        get_entities_by_position(next_state, agent.x
        , agent.y)
        items_at_front_location =
        get_entities_by_position(next_state,
        front_x, front_y)
        pickable_items = [item for item in
        items_at_agent_location +
        items_at_front_location if item.name not in
        ['Door','Wall', 'Agent']]
        if pickable_items:
            agent.carrying = pickable_items[0]
            next_state.remove(pickable_items[0])
    ...
```

**Guided by imagined transit-func**
**Trying to toggle the door**

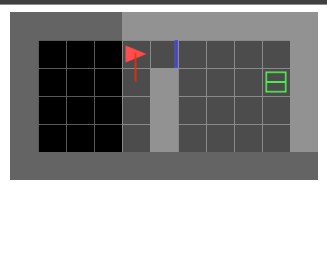

**Data-consistent transit-func**
**(correct some detail about opening the door)**

```
...
def toggle_door(agent, next_state, next_x, next_y):
    doors_in_next_position = [door for door in
    get_entities_by_position(next_state, next_x,
    next_y) if door.name == 'Door']
    if doors_in_next_position and
    doors_in_next_position[0].color == agent.
    carrying.color :
        doors_in_next_position[0].state = 'open'
...
```

<table>
<tr><td>

**Guided by imagined transit-func**
Trying to pickup the box while carrying the key

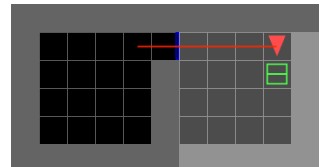

</td><td>

**Goal-driven transit-func**
(figured out cannot pickup objects while carrying others; improve the drop action that was imagined before) (SUCCEED!!)

```
...
def drop_item(agent, next_state, next_x, next_y):
    entities_in_next_position =
    get_entities_by_position(next_state, next_x,
    next_y)
    if not entities_in_next_position and agent.
    carrying:
        # Drop can only drop object if there's no
        obstacle and agent carries something.
        agent.carrying.x, agent.carrying.y = next_x,
         next_y
        next_state.append(agent.carrying)
        agent.carrying = None
...
def check_no_obstacle_between(agent, next_state, x,
y):
    dx, dy = x - agent.x, y - agent.y
    for i in range(min(abs(dx), abs(dy))):
        entities_at_next_position =
        get_entities_by_position(next_state, agent.x
         + i * dx, agent.y + i * dy)
        if any(isinstance(entity, Wall) or (
        isinstance(entity, Door) and entity.state !=
         'open') for entity in
        entities_at_next_position):
            return False
    return True
def pickup_item(agent, next_state):
    items_in_current_location =
    get_entities_by_position(next_state, agent.x,
    agent.y)
    pickable_items = [item for item in
    items_in_current_location if item.name not in ['
    Door', 'Wall', 'Agent']]
    if agent.carrying is None: # Agent can only pick
    up an item when it is not carrying an item
        if not pickable_items:
            dx, dy = agent.direction
            facing_x, facing_y = agent.x + dx, agent
            .y + dy
            if check_no_obstacle_between(agent,
            next_state, facing_x, facing_y):  # Make
             sure there is no wall or door between
            the agent and the item
                items_in_facing_location =
                get_entities_by_position(next_state,
                 facing_x, facing_y)
                pickable_items = [item for item in
                items_in_facing_location if item.
                name not in ['Door', 'Wall']]
        if pickable_items:
            agent.carrying = pickable_items[0]
            next_state.remove(pickable_items[0])
...
```

</td></tr>
</table>

**The final synthesized world model code**

```
class Entity:
```

```python
    def __init__(self, x, y, **kwargs):
        self.name = self.__class__.__name__
        self.x = x
        self.y = y
        for key, value in kwargs.items():
            setattr(self, key, value)
    def __repr__(self):
        attr = ', '.join(f'{key}={value}' for key, value in self.__dict__.items
        () if key not in ('name', 'x', 'y'))
        if attr: return f"{self.name}({self.x}, {self.y}, {attr})"
        else: return f"{self.name}({self.x}, {self.y})"
    def __eq__(self, other):
        return all(getattr(self, key) == getattr(other, key, None) for key in
        self.__dict__.keys())
    def __hash__(self):
        return hash(tuple(sorted(self.__dict__.items())))
class Agent(Entity): pass
class Key(Entity): pass
class Door(Entity): pass
class Goal(Entity): pass
class Wall(Entity): pass
class Box(Entity): pass
class Ball(Entity): pass
class Lava(Entity): pass
def update_direction(agent, action):
    all_directions = [(0, -1), (-1, 0), (0, 1), (1, 0)]
    current_dir_idx = all_directions.index(agent.direction)
    if action == 'turn right':
        agent.direction = all_directions[(current_dir_idx - 1) % 4]
    else:  # turn left
        agent.direction = all_directions[(current_dir_idx + 1) % 4]
def drop_item(agent, next_state, next_x, next_y):
    entities_in_next_position = get_entities_by_position(next_state, next_x,
    next_y)
    if not entities_in_next_position and agent.carrying:
        # Drop can only drop object if there's no obstacle and agent carries
        something.
        agent.carrying.x, agent.carrying.y = next_x, next_y
        next_state.append(agent.carrying)
        agent.carrying = None
def toggle_door(agent, next_state, next_x, next_y):
    doors_in_next_position = [door for door in get_entities_by_position(
    next_state, next_x, next_y) if door.name == 'Door']
    if doors_in_next_position and doors_in_next_position[0].color == agent.
    carrying.color :
        doors_in_next_position[0].state = 'open'
def get_entities_by_name(entities, name):
    return [ entity for entity in entities if entity.name == name ]
def get_entities_by_position(entities, x, y):
    return [ entity for entity in entities if entity.x == x and entity.y == y ]
def check_no_obstacle_between(agent, next_state, x, y):
    dx, dy = x - agent.x, y - agent.y
    for i in range(min(abs(dx), abs(dy))):
        entities_at_next_position = get_entities_by_position(next_state, agent.
        x + i * dx, agent.y + i * dy)
        if any(isinstance(entity, Wall) or (isinstance(entity, Door) and entity
        .state != 'open') for entity in entities_at_next_position):
            return False
    return True
def pickup_item(agent, next_state):
    items_in_current_location = get_entities_by_position(next_state, agent.x,
    agent.y)
    pickable_items = [item for item in items_in_current_location if item.name
    not in ['Door', 'Wall', 'Agent']]
```

```python
        if agent.carrying is None:  # Agent can only pick up an item when it is not
        carrying an item
            if not pickable_items:
                dx, dy = agent.direction
                facing_x, facing_y = agent.x + dx, agent.y + dy
                if check_no_obstacle_between(agent, next_state, facing_x, facing_y)
                    :  # Make sure there is no wall or door between the agent and the
                    item
                    items_in_facing_location = get_entities_by_position(next_state,
                     facing_x, facing_y)
                    pickable_items = [item for item in items_in_facing_location if
                    item.name not in ['Door', 'Wall']]
            if pickable_items:
                agent.carrying = pickable_items[0]
                next_state.remove(pickable_items[0])
def transition(state, action):
    next_state = list(state)
    agent = get_entities_by_name(next_state, 'Agent')[0]
    dx, dy = agent.direction
    front_x, front_y = agent.x + dx, agent.y + dy
    if action == 'turn right' or action == 'turn left':
        update_direction(agent, action)
    elif action == 'move forward':
        update_position(agent, next_state, front_x, front_y)
    elif action == 'pick up':
        pickup_item(agent, next_state)
    elif action == 'drop':
        drop_item(agent, next_state, front_x, front_y)
    elif action == 'toggle':
        toggle_door(agent, next_state, front_x, front_y)
    return next_state
def update_position(agent, next_state, next_x, next_y):
    entities_at_next_position = get_entities_by_position(next_state, next_x,
    next_y)
    if not any(
        (
            isinstance(entity, Wall) or
            isinstance(entity, Box) or
            isinstance(entity, Ball) or
            isinstance(entity, Lava) or
            (isinstance(entity, Door) and entity.state != 'open') or
            isinstance(entity, Key)
        )
        for entity in entities_at_next_position
    ):
        agent.x, agent.y = next_x, next_y
    else:
        agent.x, agent.y = agent.x, agent.y  # Agent stays in place
```

## B  Planning with Monte Carlo Tree Search and the BM25-based Heuristic

We propose a BM25-based heuristic to guide planning in text-based environments, and trade off the exploration and exploitation using a variant of Monte Carlo Tree Search (MCTS) for deterministic environments with sparse rewards. The heuristic encourages the planner to focus on trajectories that are "closer" to the goal as specified by users, while MCTS enables the planner to explore other options after failing on the seemingly promising ones for too long.

### B.1  Monte Carlo Tree Search for Deterministic Environments with Sparse Rewards

Monte Carlo Tree Search (MCTS) [12, 5] is a classic heuristic search algorithm and has achieved speculative success in difficult problems such as *Go* [58]. It uses tree policies such as UCT (Upper

---
**Algorithm 2** Monte Carlo Tree Search for Deterministic Sparse-Reward Environment
---

**function** UCTSEARCH($s_0$)                                    ▷ MCTS Search with the UCT tree policy
    create the root node $v_0$ with state $s_0$
    **while** within computation budget **do**
        $v_l \leftarrow$ TREEPOLICY($v_0$)
        $v_l' \leftarrow$ EXPAND($v_l$)
        $\Delta \leftarrow h(v_l')$                      ▷ *No more rollouts, BM25-based heuristic*
        BACKUP($v_l'$)
    **return** $a$(BESTCHILD($v_0$, 0))
**function** TREEPOLICY($v$)                          ▷ Select node to expand using the UCT criterion
    **while** $v$ is nonterminal **do**
        **if** $v$ not fully expanded **then**
            **return** EXPAND($v$)
        **else**
            $v \leftarrow$ BESTCHILD($v$, $C$)
    **return** $v$
**function** EXPAND($v$)                                        ▷ Select a child to expand
    choose $a \in$ untried actions from $A(s(v))$
    add a new child $v'$ to $v$ with $s(v'), r(v') = $ WorldModel($s(v), a$) and $a(v') = a$
    **return** $v'$
**function** BESTCHILD($v$, $c$)                          ▷ The UCT criterion for selecting nodes
    **return** $\arg\max_{v' \in \text{children of } v} \frac{Q(v')}{N(v')} + c\sqrt{\frac{2\ln N(v)}{N(v')}}$ ▷ Exploit: $\frac{Q(v')}{N(v')}$, Exploration: $c\sqrt{\frac{2\ln N(v)}{N(v')}}$
**function** BACKUP($v$, $\Delta$)                        ▷ Back propagation of the heuristic values
    **while** $v$ is not `null` **do**
        $N(v) \leftarrow N(v) + 1$
        $Q(v) \leftarrow Q(v) + \Delta$
        $v \leftarrow$ parent of $v$

---

Confidence bounds for Trees) [35, 2] to trade exploration and exploitation while deciding on which part of the search space to search more. It then evaluates the potential benefits of searching in this selected part of space, i.e., trajectories with the same prefix as the selected partial trajectory, by rolling them out in simulations. The UCT tree policy makes a balance between exploiting the search space with higher potential benefits and exploring the lesser-considered regions.

However, the MCTS algorithm is designed for general planning problems and is not efficient enough for deterministic environments with sparse rewards, such as Alfworld in our experiments. We thus modify the algorithm to make it faster by making use of the additional assumptions of the environments: deterministic and sparse rewards. Specifically, we remove the rollout procedure and instead substitute it with an optional heuristic to estimate the promise of the partial trajectories. Noticing that there is no randomness and no partial rewards in the environments, roll-outs are not cost-effective in the usage of simulations. We do not need to run a trajectory multiple times to estimate its expectations of returns. All returns of partial trajectories are zero and, once a trajectory gets a positive return, we find the goal state and should immediately return the trajectory from the initial state to this goal state. Computation spent on simulating the roll-outs should instead be spent on expanding the search tree with the smarter online tree policy rather than the fixed one in roll-outs. We thus remove the rollout procedure, or equivalently, set the rollout depth to zero with a hard-coded Q-value estimator, i.e., the heuristic. We adopt the classic UCT-based MCTS algorithm and replace the rollouts with a BM25-based heuristic in our experiments, as shown in Algorithm 2. We refer the readers to the other surveys [5] for details about MCTS.

## B.2 BM25 as a Heuristic for Planning in Text-based Environments

We propose to estimate the promise/goodness a selected state with information on the similarity between the goal (specified by the user in texts) and the partial trajectory (texts that describe the path the agent has traversed from the initial state to the selected state). An example of the goal and the partial trajectory in the Alfworld environment is as follows:

```
Mission example:
  put a alarmclock in desk

Partial trajectory example 1:
  Goto(dest=sidetable1)
  The agent1 at loc5 is now at loc20.
  Pickup(obj=alarmclock1, receptacle=sidetable1)
  The agent1 is now holding alarmclock1. The alarmclock1 at loc20 is now at None
  Goto(dest=desk1)
  The agent1 at loc20 is now at loc1.

Partial trajectory example 2:
  Goto(dest=microwave1)
  The agent1 at loc41 is now at loc36.
  Heat(obj=apple1, receptacle=microwave1)
  The apple1 at None is now hot.
  Goto(dest=garbagecan1)
  The agent1 at loc36 is now at loc41.
```

Intuitively, the assumption is that the more similar the partial trajectory is to the goal, the more relevant the current state is to the goal, and thus the closer the current state is to the goal states. For example, if the goal is "put a hot alarmclock in desk", trajectories that involve "alarmclock", "desk", and "hot" should be more promising to exploit than other irrelevant ones.

There are various metrics and methods from the text retrieval field to estimate the text similarities [43, 34, 33, 80]. We focus on the symbolic ones for efficiency issues and find BM25 [50], a state-of-the-art algorithm for web-scale information retrieval, works significantly well in our experiments. Our algorithm is different from the classic BM25 algorithm in the sense that we do not have a static previously collected trajectory/document corpus. Instead, we maintain an online trajectory corpus during the search, as shown in Algorithm 3 and Algorithm 2.

One interesting finding in our preliminary experiments is that the saturating formulation of the term frequency, $\frac{n_\tau \cdot (k_1+1)}{n_\tau + k_1 \cdot (1-b+b \cdot |\tau|/l_D)}$, in BM25 significantly outperforms its linear variant in TF-IDF. Without the saturating formulation, the heuristic will assign too high scores to seemingly promising too long trajectories such as those that repetitively pick up "alarmclock" and go to "desk" (the concrete states are different in minor irrelevant details throughout the trajectory, so they are not the same), which results in the planner to be stuck in meaningless local minimum until exhausting the budget.

## C   More Experimental Results

### C.1   PPO for MiniGrid

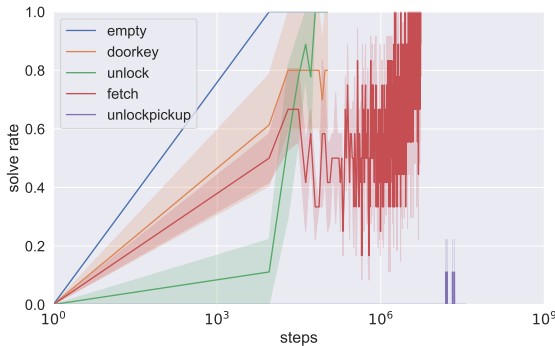

Figure 6: Performance of PPO in MiniGrid environments.

---

**Algorithm 3** Online BM25-based Heuristic

---

**Hyperparam:** $k_1$ (default to $1.5$)
**Hyperparam:** $b$ (default to $0.75$)

$N \leftarrow 0$               $\triangleright$ Number of Trajectories
$Nt(\cdot) \leftarrow 0$         $\triangleright$ Number of trajectories for each token
$l_D \leftarrow 0$          $\triangleright$ Average length of trajectories
**function** BM25HEURISTIC(node $v$, mission $m$)
    **global** $N, Nt, l_D$
    $\tau \leftarrow$ TRAJECTORY($v$)    $\triangleright$ Trajectory from the root $v_0$ to the node $v$, in text as a list of tokens
    $m \leftarrow$ PARSE($m$)          $\triangleright$ Parse mission in text into a list of tokens
    **loop** $t \in$ SET($\tau$)         $\triangleright$ Update the number of trajectories for each token
        $Nt(t) \leftarrow Nt(t) + 1$
    $l_D \leftarrow \frac{l_D \times N + |\tau|}{N+1}$         $\triangleright$ Update the average length of trajectories
    $N \leftarrow N + 1$         $\triangleright$ Update the total number of trajectories
    $h \leftarrow 0$         $\triangleright$ Initialize the heuristic value
    **loop** $t \in$ SET($m$)
        $n_\tau \leftarrow$ COUNT($\tau, t$)         $\triangleright$ Number of token $t$ in trajectory $\tau$
        $n_m \leftarrow$ COUNT($m, t$)         $\triangleright$ Number of token $t$ in mission $m$
        $idf \leftarrow \ln\left(\frac{N - Nt(t) + 0.5}{Nt(t) + 0.5} + 1\right)$         $\triangleright$ Inverse document frequency
        $h \leftarrow h + \frac{n_m}{|m|} \cdot idf \cdot \frac{n_\tau \cdot (k_1+1)}{n_\tau + k_1 \cdot (1 - b + b \cdot |\tau|/l_D)}$         $\triangleright$ BM25 Score
    **return** $h$

---

We evaluate PPO, as a Deep RL baseline, in minigrid experiments. We use the tuned hyper-parameters from RL Baselines3 Zoo [48] for each environment and use the same symbolic memory state as ours. As shown in Figure 6, PPO is much more sample inefficient than ours. It needs $10^4 - 10^5$ steps to learn valid policies (not perfect though) in most environments. It cannot solve the UnlockPickup environment even in $3 \times 10^8$ steps.

## C.2 PPO for Sokoban

The PPO baseline was implemented using Stable Baselines3 [49] and the gym-sokoban library [54] with 256 parallel environments, a batch_size of 2048, a horizon of 50 steps, and the rest default hyperparameters according to the Stable Baselines3 library (n_steps 2048, learning rate 0.0003, gamma 0.99, n_epochs 10, gae_lambda 0.95, clip_range 0.2, normalized advantage, ent_coef 0, vf_coef 0.5).

The input to the network is a $(3, 7, 7)$ corresponding to the 7 by 7 Sokoban grid, with each item in the grid represent as an RGB pixel.

The policy network used was a convolutional neural network with 3 convolutional layers with 16, 32, and 64 filters and all with a kernel of size $(2, 2)$ and a stride of $1$, followed by a linear layer with 9 outputs, corresponding to the 9 possible actions. All layers are separated by Rectified Linear Units (ReLu).

## C.3 DreamerV3 for Sokoban

The DreamerV3 baseline was implemented using the simple training script provided by the open-source DreamerV3 library created by Danijar Hafner [27], and the gym-sokoban library [54] with default hyperparameters and 1 environment.

# D  Example of synthesized world models for Sokoban

**Synthesized transition function for Sokoban**

```python
def transition(state, action):
    """
    Args:
        state: a set of entities representing the state of the environment
        action: the action can be "move right", "move left", "move up", "
        move down"
    Returns:
        next_state: the next state of the environment
    """
    # here we define how the player coordinates change for each action
    action_to_delta = {
        "move right": (1, 0),
        "move left": (-1, 0),
        "move up": (0, -1),
        "move down": (0, 1)
    }
    # Here we get the player and the boxes in the current state
    player = get_entities_by_name(state, 'Player')[0]
    boxes = get_entities_by_name(state, 'Box')
    walls = get_entities_by_name(state, 'Wall')
    # Then, we calculate the new player position according to the action
    delta_x, delta_y = action_to_delta[action]
    new_player_x = player.x + delta_x
    new_player_y = player.y + delta_y
    # We check if the new player position is a Wall
    if get_entities_by_position(walls, new_player_x, new_player_y):
        # If so, the player does not move
        pass
    else:
        # If not, the player moves to the new position
        pushed_box = get_entities_by_position(boxes, new_player_x,
        new_player_y)
        if pushed_box:
            pushed_box_x = pushed_box[0].x + delta_x
            pushed_box_y = pushed_box[0].y + delta_y
            # Check if there is a wall or other box at the pushed box
            destination
            if get_entities_by_position(boxes + walls, pushed_box_x,
            pushed_box_y):
                # If so, the player and the box do not move
                pass
            else:
                # If not, the box moves to the new position
                pushed_box[0].x += delta_x
                pushed_box[0].y += delta_y
                player.x += delta_x
                player.y += delta_y
        else:
            player.x += delta_x
            player.y += delta_y
    return state
```

**Synthesized reward function for Sokoban**

```python
def reward_func(state, action, next_state):
    reward = -0.1
```

```python
        done = False
        boxes_prev = get_entities_by_name(state, 'Box')
        targets_prev = get_entities_by_name(state, 'Target')
        boxes_next = get_entities_by_name(next_state, 'Box')
        targets_next = get_entities_by_name(next_state, 'Target')
        for box in boxes_next:
            if any(box.x == target.x and box.y == target.y for target in
            targets_next):
                if not any(box.x == prev_box.x and box.y == prev_box.y for
                prev_box in boxes_prev
                            if any(prev_box.x == prev_target.x and prev_box.y ==
                            prev_target.y for prev_target in targets_prev)):
                    reward += 1
        for box in boxes_prev:
            if any(box.x == target.x and box.y == target.y for target in
            targets_prev):
                if not any(box.x == next_box.x and box.y == next_box.y for
                next_box in boxes_next
                            if any(next_box.x == next_target.x and next_box.y ==
                            next_target.y for next_target in targets_next)):
                    reward -= 1
        if all(any(box.x == target.x and box.y == target.y for target in
        targets_next) for box in boxes_next):
            reward += 10
            done = True
        return reward, done
```

## E   Example of synthesized world model for Alfworld

Synthesized transition function for Alfworld

```python
class Entity:
    def __init__(self, name, loc=None, in_on=None, **kwargs):
        self.name = name
        self.loc = loc
        self.in_on = in_on
        self.ishot, self.iscool, self.isclean = None, None, None
        self.isopen, self.ison, self.istoggled = None, None, None
        self.pickupable, self.openable, self.toggleable = None, None, None
        self.heatable, self.coolable, self.cleanable = None, None, None
        self.isobject, self.isreceptacle, self.isreceptacleobject = None,
        None, None
        for key, value in kwargs.items():
            setattr(self, key, value)
    def __repr__(self):
        attr_list = []
        for key in self.__dict__.keys():
            if getattr(self, key) is not None:
                value = getattr(self, key)
                if isinstance(value, str):
                    value = f'"{value}"'
                attr_list.append(f"{key}={value}")
        attr = ', '.join(attr_list)
        class_name = self.entity_type().capitalize()
        return f"{class_name}({attr})"
    def __eq__(self, other):
        return all(getattr(self, key) == getattr(other, key, None) for key
        in self.__dict__.keys())
    def __hash__(self):
```

```python
            return hash(tuple(sorted(self.__dict__.items())))
    def entity_type(self):
        return entity_type(self.name)
    def is_entity_type(self, etype):
        return is_entity_type(self.name, etype)
class Agent(Entity):
    def __init__(self, name, loc=None, holding=None, **kwargs):
        super().__init__(name, loc, **kwargs)
        self.holding = holding
    def __repr__(self):
        return f"{self.__class__.__name__}(name={self.name}, loc={self.loc},
         holding={self.holding})"
class Action:
    def __init__(self, name, **kwargs,):
        for key, value in kwargs.items():
            setattr(self, key, value)
    def __repr__(self):
        attr = ', '.join([f"{key}={value}" for key, value in self.__dict__.
        items()])
        return f"{self.__class__.__name__}({attr})"
    def __eq__(self, other):
        return all(getattr(self, key) == getattr(other, key, None) for key
        in self.__dict__.keys())
    def __hash__(self):
        return hash(tuple(sorted(self.__dict__.items())))
class Goto(Action):
    def __init__(self, entity_name,):
        super().__init__('goto', dest=entity_name)
class Open(Action):
    def __init__(self, entity_name,):
        super().__init__('open', obj=entity_name)
class Close(Action):
    def __init__(self, entity_name,):
        super().__init__('close', obj=entity_name)
class Examine(Action):
    def __init__(self, entity_name,):
        super().__init__('examine', obj=entity_name)
class Use(Action):
    def __init__(self, entity_name,):
        super().__init__('use', obj=entity_name)
class Pickup(Action):
    def __init__(self, entity1_name, entity2_name):
        super().__init__('pickup', obj=entity1_name, receptacle=entity2_name
        )
class Put(Action):
    def __init__(self, entity1_name, entity2_name, cancontain,):
        super().__init__('put', obj=entity1_name, receptacle=entity2_name,
        cancontain=cancontain)
class Clean(Action):
    def __init__(self, entity1_name, entity2_name):
        super().__init__('clean', obj=entity1_name, receptacle=entity2_name)
class Heat(Action):
    def __init__(self, entity1_name, entity2_name):
        super().__init__('heat', obj=entity1_name, receptacle=entity2_name)
class Cool(Action):
    def __init__(self, entity1_name, entity2_name):
        super().__init__('cool', obj=entity1_name, receptacle=entity2_name)
def entity_type(name):
    return name.strip('0123456789').strip()
def is_entity_type(name, etype):
    return etype.lower().strip() == entity_type(name).lower().strip()
def is_receptacle_open(state, obj):
    if obj.in_on is None:
```

```python
            return True
    receptacle = get_entity_by_name(state, obj.in_on)
    if receptacle is None or not receptacle.openable:
        return True
    return receptacle.isopen
def move_and_heat_with_microwave(state, action_obj_name,
action_receptacle_name):
    agent = get_entities_by_type(state, 'agent')[0]
    heating_entity = get_entity_by_name(state, action_obj_name)
    microwave_entity = get_entity_by_name(state, action_receptacle_name)
    # Ensure agent is in the correct location where the microwave is
    if agent.loc != microwave_entity.loc:
        agent.loc = microwave_entity.loc
    # Checking whether the agent has the object
    if agent.holding == heating_entity.name and heating_entity.heatable:
        # Heat the object
        heating_entity.ishot = True
        heating_entity.iscool = False   # Resetting cool
    return state
#create_agent() function creates an agent entity instance
def create_agent(name, loc=None, holding=None, **kwargs):
    return Agent(name, loc=loc, holding=holding, **kwargs)
# Correcting and optimizing the Python code to better simulate world logic
# Importing necessary libraries
from copy import deepcopy
def get_entities_by_type(entities, etype):
    return [entity for entity in entities if entity.entity_type().lower() ==
     etype.lower()]
def is_receptacle(entity):
    return getattr(entity, 'isreceptacle', False)
def move_and_heat(state, obj_name, receptacle_name):
    # Similar implementation as before is assumed
    pass
def get_entity_by_type(entities, type_name):
    return next((entity for entity in entities if entity.entity_type().lower
    () == type_name.lower()), None)
def get_entity_by_name(entities, name):
    return next((entity for entity in entities if _canonicalize_name(entity.
    name) == _canonicalize_name(name)), None)
def _canonicalize_name(name):
    return ''.join(name.split()).lower()
def open_state(state, obj_name):
    obj = get_entity_by_name(state, obj_name)
    if obj and obj.openable and not obj.isopen:
        obj.isopen = True
    return state
def close_state(state, obj_name):
    obj = get_entity_by_name(state, obj_name)
    if obj and obj.openable and obj.isopen:
        obj.isopen = False
    return state
def move_and_put(state, obj_name, receptacle_name):
    agent = get_entity_by_type(state, 'Agent')
    obj = get_entity_by_name(state, obj_name)
    receptacle = get_entity_by_name(state, receptacle_name)
    if obj and receptacle and obj.loc == agent.loc and agent.holding ==
    obj_name:
        if receptacle.openable and not receptacle.isopen:
            return state  # Can't put into a closed receptacle
        obj.loc = receptacle.loc
        obj.in_on = receptacle.name
        agent.holding = None
    return state
```

```python
def set_agent_holding(state, agent_name, obj_name):
    """ Sets the agent's holding attribute to the object's name if possible.
    """
    agent = get_entity_by_name(state, agent_name)
    obj = get_entity_by_name(state, obj_name)
    if obj.pickupable:
        agent.holding = obj_name
        obj.in_on = None  # The object is no longer in or on a receptacle
        obj.loc = None
def move_and_pickup(state, obj_name):
    """ Moves the agent to the object's location and then picks it up, if
    near and pickupable """
    agent = get_entity_by_type(state, 'Agent')
    obj = get_entity_by_name(state, obj_name)
    if obj and obj.pickupable and obj.loc == agent.loc:
        set_agent_holding(state, agent.name, obj.name)
    return state
# Assumed helper functions (simplified for clarity)
def move_and_use(state, obj_name):
    # Implement action logic here if needed
    pass
# Example function signatures, defining other required actions similarly
def move_agent(state, destination_name):
    agent = get_entity_by_type(state, 'Agent')
    destination = get_entity_by_name(state, destination_name)
    if destination:
        agent.loc = destination.loc
    return state
def open_receptacle(state, receptacle_name):
    receptacle = get_entity_by_name(state, receptacle_name)
    if receptacle and receptacle.openable and not receptacle.isopen:
        receptacle.isopen = True
    return state
def create_item(name, loc=None, in_on=None, **kwargs):
    return Entity(name, loc=loc, in_on=in_on, **kwargs)
def cool_object(state, obj_name, receptacle_name):
    """
    Refactored to ensure the receptacle is open before placing the object
    inside.
    """
    receptacle = get_entity_by_name(state, receptacle_name)
    if receptacle.openable and not is_open(receptacle):
        state = open_receptacle(state, receptacle_name)
    return apply_cooling(state, obj_name, receptacle_name)
# Helper Function
def is_open(entity):
    return getattr(entity, 'isopen', False)
# Helper function to ensure a receptacle is open and updated correctly
def open_receptacle_if_closed(state, receptacle_name):
    receptacle = get_entity_by_name(state, receptacle_name)
    if receptacle and receptacle.openable and not receptacle.isopen:
        receptacle.isopen = True
    return state
def apply_cooling(state, obj_name, receptacle_name):
    agent = get_entity_by_name(state, 'Agent')
    obj = get_entity_by_name(state, obj_name)
    receptacle = get_entity_by_name(state, receptacle_name)
    if obj and receptacle and obj.coolable:
        if agent.holding == obj.name or (obj.in_on == receptacle.name and
        is_open(receptacle)):
            if not is_open(receptacle):
                state = open_receptacle_if_closed(state, receptacle_name)
            obj.loc = None
```

```python
                obj.in_on = receptacle_name
                obj.iscool = True
                obj.ishot = False
                agent.holding = None
            else:
                print("Agent does not hold the object or the fridge is not open.
                ")
        return state
    def transition(state, action):
        """
        Perform a transition based on the action type and update the state
        accordingly.
        Args:
            state (set): Current state represented as a set of entities.
            action (Action): Action to be executed.
        Returns:
            set: Updated state after the action has been performed.
        """
        state = deepcopy(state)  # Copy state to avoid mutation
        action_type = type(action).__name__.lower()
        if action_type == "goto":
            return move_agent(state, action.dest)
        elif action_type == "open":
            return open_receptacle(state, action.obj)
        elif action_type == "close":
            return close_state(state, action.obj)
        elif action_type == "pickup":
            return move_and_pickup(state, action.obj)
        elif action_type == "put":
            return move_and_put(state, action.obj, action.receptacle)
        elif action_type == "use":
            return move_and_use(state, action.obj)
        elif action_type == "heat":
            return move_and_heat(state, action.obj, action.receptacle)
        elif action_type == "cool":
            return move_and_cool(state, action.obj, action.receptacle)
        else:
            raise ValueError(f"Unsupported action type: {action_type}")
        return state
    def move_and_cool(state, obj_name, receptacle_name):
        """
        Move an object to a cooling receptacle (e.g., a fridge) and cool it down
        .
        Args:
            state (set): The current state of the environment.
            obj_name (str): The name of the object to be cooled.
            receptacle_name (str): The name of the receptacle where the object
            will be cooled.
        Returns:
            set: The updated state after the cooling process.
        """
        agent = get_entity_by_name(state, 'Agent')
        obj = get_entity_by_name(state, obj_name)
        receptacle = get_entity_by_name(state, receptacle_name)
        if obj and receptacle and obj.coolable:
            if agent.holding == obj_name:
                if receptacle.openable and not receptacle.isopen:
                    state = open_receptacle(state, receptacle.name)
                agent.holding = None  # Agent releases the object
                obj.in_on = receptacle.name  # Place object in/on the receptacle
                obj.loc = None  # Since it's inside the receptacle, location is
                not direct
                obj.iscool = True
```

```
            obj.ishot = False
        else:
            print("Condition not met: Agent must hold the object.")
    else:
        print("Condition not met: Object must be coolable and receptacle
        must exist.")
    return state
```

Example reward function:

**Synthesized reward function for "put a hot egg in fridge"**

```
# put a hot egg in fridge
def get_entities_by_type(entities, etype):
    return [ entity for entity in entities if entity.is_entity_type(etype) ]
def get_entities_by_loc(entities, loc):
    entities = [ entity for entity in entities if entity.loc == loc ]
    return entities
def entity_type(name):
    return name.strip('0123456789').strip()
def is_entity_type(name, etype):
    return etype.lower().strip() == entity_type(name).lower().strip()
def reward_func(state, action, next_state):
    """
    Args:
        state: the state of the environment
        action: the action to be executed
        next_state: the next state of the environment
    Returns:
        reward: the reward of the action
        done: whether the episode is done
    """
    agent = next(iter(get_entities_by_type(state, 'agent')))
    next_agent = next(iter(get_entities_by_type(next_state, 'agent')))
    # If it's a put action, check if the moving object is egg and
    destination is fridge
    if isinstance(action, Put):
        obj_name = action.obj
        rec_name = action.receptacle
        if is_entity_type(obj_name, 'egg') and is_entity_type(rec_name, '
        fridge') and not next_agent.holding:
            matching_entity = [entity for entity in get_entities_by_loc(
            next_state, agent.loc) if entity.name == obj_name and entity.
            in_on == rec_name]
            if matching_entity:
                # Check if the egg is hot
                if matching_entity[0].ishot:
                    return 1, True
                else:
                    return 0, False
            else:
                return 0, False
    # Other parts of the reward function
    # If the action is not a put action or the object is not a egg or the
    destination is not a fridge, return 0 reward and done=False
    return 0, False
```

# F Prompts

We list all the prompts that are used in our experiments in this section. The functionality for each prompt is stated in its subsection name. We highlight the dynamic information as yellow and the main instruction as blue. The dynamic information includes the data collected so far in the replay buffer and the codes synthesized so far by previous LLM calls.

## F.1 Initializing the transition function

It asks LLMs to generate a transition function $(s, a) \rightarrow s'$ following the code template to model seven uniformly randomly sampled experience data $(s, a, s')$ in the replay buffer.

```
You are a robot exploring in an object-centric environment. Your goal is to
model the logic of the world in python. You will be provided experiences in
the format of (state, action, next_state) tuples. You will also be provided
with a short natural language description that briefly summarizes the
difference between the state and the next state for each (state, next_state
,) pair. You need to implement the python code to model the logic of the
world, as seen in the provided experiences. Please follow the template to
implement the code. The code needs to be directly runnable on the state and
return the next state in python as provided in the experiences.

You need to implement python code to model the logic of the world as seen in
 the following experiences:

The action "toggle" transforms the state from
```
Wall(0, 0) ;    Wall(1, 0) ;    Wall(2, 0) ;    Wall(3, 0) ;    Wall(4, 0) ;
Wall(0, 1) ;    empty ;         Wall(2, 1) ;    empty ;         Wall(4, 1) ;
Wall(0, 2) ;    Agent(1, 2, direction=(0, -1), carrying=None) ; Door(2, 2,
color=yellow, state=locked) ;        empty ;         Wall(4, 2) ;
Wall(0, 3) ;    Key(1, 3, color=yellow) ;       Wall(2, 3) ;    Goal(3, 3) ;
    Wall(4, 3) ;
Wall(0, 4) ;    Wall(1, 4) ;    Wall(2, 4) ;    Wall(3, 4) ;    Wall(4, 4) ;
```
to
```
Wall(0, 0) ;    Wall(1, 0) ;    Wall(2, 0) ;    Wall(3, 0) ;    Wall(4, 0) ;
Wall(0, 1) ;    empty ;         Wall(2, 1) ;    empty ;         Wall(4, 1) ;
Wall(0, 2) ;    Agent(1, 2, direction=(0, -1), carrying=None) ; Door(2, 2,
color=yellow, state=locked) ;        empty ;         Wall(4, 2) ;
Wall(0, 3) ;    Key(1, 3, color=yellow) ;       Wall(2, 3) ;    Goal(3, 3) ;
    Wall(4, 3) ;
Wall(0, 4) ;    Wall(1, 4) ;    Wall(2, 4) ;    Wall(3, 4) ;    Wall(4, 4) ;
```
The difference is
"""
Nothing happened
"""

The action "toggle" transforms the state from
```
Wall(0, 0) ;    Wall(1, 0) ;    Wall(2, 0) ;    Wall(3, 0) ;    Wall(4, 0) ;
Wall(0, 1) ;    empty ;         Wall(2, 1) ;    empty ;         Wall(4, 1) ;
Wall(0, 2) ;    Agent(1, 2, direction=(0, 1), carrying=None) ;  Door(2, 2,
color=yellow, state=locked) ;        empty ;         Wall(4, 2) ;
Wall(0, 3) ;    Key(1, 3, color=yellow) ;       Wall(2, 3) ;    Goal(3, 3) ;
    Wall(4, 3) ;
Wall(0, 4) ;    Wall(1, 4) ;    Wall(2, 4) ;    Wall(3, 4) ;    Wall(4, 4) ;
```
to
```

```
Wall(0, 0) ;    Wall(1, 0) ;    Wall(2, 0) ;    Wall(3, 0) ;    Wall(4, 0) ;
Wall(0, 1) ;    empty ;         Wall(2, 1) ;    empty ;         Wall(4, 1) ;
Wall(0, 2) ;    Agent(1, 2, direction=(0, 1), carrying=None) ; Door(2, 2,
color=yellow, state=locked) ;        empty ;        Wall(4, 2) ;
Wall(0, 3) ;    Key(1, 3, color=yellow) ;       Wall(2, 3) ;    Goal(3, 3) ;
    Wall(4, 3) ;
Wall(0, 4) ;    Wall(1, 4) ;    Wall(2, 4) ;    Wall(3, 4) ;    Wall(4, 4) ;
```
The difference is
"""
Nothing happened
"""

The action "turn right" transforms the state from
```
Wall(0, 0) ;    Wall(1, 0) ;    Wall(2, 0) ;    Wall(3, 0) ;    Wall(4, 0) ;
Wall(0, 1) ;    empty ;         Wall(2, 1) ;    empty ;         Wall(4, 1) ;
Wall(0, 2) ;    Agent(1, 2, direction=(1, 0), carrying=None) ; Door(2, 2,
color=yellow, state=locked) ;        empty ;        Wall(4, 2) ;
Wall(0, 3) ;    Key(1, 3, color=yellow) ;       Wall(2, 3) ;    Goal(3, 3) ;
    Wall(4, 3) ;
Wall(0, 4) ;    Wall(1, 4) ;    Wall(2, 4) ;    Wall(3, 4) ;    Wall(4, 4) ;
```
to
```
Wall(0, 0) ;    Wall(1, 0) ;    Wall(2, 0) ;    Wall(3, 0) ;    Wall(4, 0) ;
Wall(0, 1) ;    empty ;         Wall(2, 1) ;    empty ;         Wall(4, 1) ;
Wall(0, 2) ;    Agent(1, 2, direction=(0, 1), carrying=None) ; Door(2, 2,
color=yellow, state=locked) ;        empty ;        Wall(4, 2) ;
Wall(0, 3) ;    Key(1, 3, color=yellow) ;       Wall(2, 3) ;    Goal(3, 3) ;
    Wall(4, 3) ;
Wall(0, 4) ;    Wall(1, 4) ;    Wall(2, 4) ;    Wall(3, 4) ;    Wall(4, 4) ;
```
The difference is
"""
The agent (direction=(1, 0)) at pos (1, 2) becomes an agent (direction=(0,
1)).
"""

The action "turn left" transforms the state from
```
Wall(0, 0) ;    Wall(1, 0) ;    Wall(2, 0) ;    Wall(3, 0) ;    Wall(4, 0) ;
Wall(0, 1) ;    empty ;         Wall(2, 1) ;    empty ;         Wall(4, 1) ;
Wall(0, 2) ;    Agent(1, 2, direction=(-1, 0), carrying=None) ; Door(2, 2,
color=yellow, state=locked) ;        empty ;        Wall(4, 2) ;
Wall(0, 3) ;    Key(1, 3, color=yellow) ;       Wall(2, 3) ;    Goal(3, 3) ;
    Wall(4, 3) ;
Wall(0, 4) ;    Wall(1, 4) ;    Wall(2, 4) ;    Wall(3, 4) ;    Wall(4, 4) ;
```
to
```
Wall(0, 0) ;    Wall(1, 0) ;    Wall(2, 0) ;    Wall(3, 0) ;    Wall(4, 0) ;
Wall(0, 1) ;    empty ;         Wall(2, 1) ;    empty ;         Wall(4, 1) ;
Wall(0, 2) ;    Agent(1, 2, direction=(0, 1), carrying=None) ; Door(2, 2,
color=yellow, state=locked) ;        empty ;        Wall(4, 2) ;
Wall(0, 3) ;    Key(1, 3, color=yellow) ;       Wall(2, 3) ;    Goal(3, 3) ;
    Wall(4, 3) ;
Wall(0, 4) ;    Wall(1, 4) ;    Wall(2, 4) ;    Wall(3, 4) ;    Wall(4, 4) ;
```
The difference is
"""
```

The agent (direction=(-1, 0)) at pos (1, 2) becomes an agent (direction=(0, 1)).
"""

The action "turn right" transforms the state from
```
Wall(0, 0) ;    Wall(1, 0) ;    Wall(2, 0) ;    Wall(3, 0) ;    Wall(4, 0) ;
Wall(0, 1) ;    empty ;         Wall(2, 1) ;    empty ;         Wall(4, 1) ;
Wall(0, 2) ;    Agent(1, 2, direction=(0, 1), carrying=None) ; Door(2, 2,
color=yellow, state=locked) ;          empty ;         Wall(4, 2) ;
Wall(0, 3) ;    Key(1, 3, color=yellow) ;       Wall(2, 3) ;    Goal(3, 3) ;
    Wall(4, 3) ;
Wall(0, 4) ;    Wall(1, 4) ;    Wall(2, 4) ;    Wall(3, 4) ;    Wall(4, 4) ;
```
to
```
Wall(0, 0) ;    Wall(1, 0) ;    Wall(2, 0) ;    Wall(3, 0) ;    Wall(4, 0) ;
Wall(0, 1) ;    empty ;         Wall(2, 1) ;    empty ;         Wall(4, 1) ;
Wall(0, 2) ;    Agent(1, 2, direction=(-1, 0), carrying=None) ; Door(2, 2,
color=yellow, state=locked) ;          empty ;          Wall(4, 2) ;
Wall(0, 3) ;    Key(1, 3, color=yellow) ;       Wall(2, 3) ;    Goal(3, 3) ;
    Wall(4, 3) ;
Wall(0, 4) ;    Wall(1, 4) ;    Wall(2, 4) ;    Wall(3, 4) ;    Wall(4, 4) ;
```
The difference is
"""
The agent (direction=(0, 1)) at pos (1, 2) becomes an agent (direction=(-1, 0)).
"""

The action "turn right" transforms the state from
```
Wall(0, 0) ;    Wall(1, 0) ;    Wall(2, 0) ;    Wall(3, 0) ;    Wall(4, 0) ;
Wall(0, 1) ;    empty ;         Wall(2, 1) ;    empty ;         Wall(4, 1) ;
Wall(0, 2) ;    Agent(1, 2, direction=(-1, 0), carrying=None) ; Door(2, 2,
color=yellow, state=locked) ;          empty ;         Wall(4, 2) ;
Wall(0, 3) ;    Key(1, 3, color=yellow) ;       Wall(2, 3) ;    Goal(3, 3) ;
    Wall(4, 3) ;
Wall(0, 4) ;    Wall(1, 4) ;    Wall(2, 4) ;    Wall(3, 4) ;    Wall(4, 4) ;
```
to
```
Wall(0, 0) ;    Wall(1, 0) ;    Wall(2, 0) ;    Wall(3, 0) ;    Wall(4, 0) ;
Wall(0, 1) ;    empty ;         Wall(2, 1) ;    empty ;         Wall(4, 1) ;
Wall(0, 2) ;    Agent(1, 2, direction=(0, -1), carrying=None) ; Door(2, 2,
color=yellow, state=locked) ;          empty ;         Wall(4, 2) ;
Wall(0, 3) ;    Key(1, 3, color=yellow) ;       Wall(2, 3) ;    Goal(3, 3) ;
    Wall(4, 3) ;
Wall(0, 4) ;    Wall(1, 4) ;    Wall(2, 4) ;    Wall(3, 4) ;    Wall(4, 4) ;
```
The difference is
"""
The agent (direction=(-1, 0)) at pos (1, 2) becomes an agent (direction=(0, -1)).
"""

The action "turn left" transforms the state from
```
Wall(0, 0) ;    Wall(1, 0) ;    Wall(2, 0) ;    Wall(3, 0) ;    Wall(4, 0) ;
Wall(0, 1) ;    empty ;         Wall(2, 1) ;    empty ;         Wall(4, 1) ;
Wall(0, 2) ;    Agent(1, 2, direction=(0, 1), carrying=None) ; Door(2, 2,
color=yellow, state=locked) ;          empty ;         Wall(4, 2) ;
```

```
Wall(0, 3) ;     Key(1, 3, color=yellow) ;        Wall(2, 3) ;    Goal(3, 3) ;
    Wall(4, 3) ;
Wall(0, 4) ;     Wall(1, 4) ;     Wall(2, 4) ;     Wall(3, 4) ;     Wall(4, 4) ;
'''
to
'''
Wall(0, 0) ;     Wall(1, 0) ;     Wall(2, 0) ;     Wall(3, 0) ;     Wall(4, 0) ;
Wall(0, 1) ;     empty ;          Wall(2, 1) ;     empty ;          Wall(4, 1) ;
Wall(0, 2) ;     Agent(1, 2, direction=(1, 0), carrying=None) ;  Door(2, 2,
color=yellow, state=locked) ;        empty ;          Wall(4, 2) ;
Wall(0, 3) ;     Key(1, 3, color=yellow) ;        Wall(2, 3) ;    Goal(3, 3) ;
    Wall(4, 3) ;
Wall(0, 4) ;     Wall(1, 4) ;     Wall(2, 4) ;     Wall(3, 4) ;     Wall(4, 4) ;
'''
The difference is
"""
The agent (direction=(0, 1)) at pos (1, 2) becomes an agent (direction=(1,
0)).
"""
```

Please implement code to model the logic of the world as demonstrated by the experiences. Here is the template for the transition function. Please implement the transition function following the template. The code needs to be directly runnable on the inputs of (state, action) and return the next state in python as provided in the experiences.

```
'''

class Entity:
    def __init__(self, x, y, **kwargs):
        self.name = self.__class__.__name__
        self.x = x
        self.y = y
        for key, value in kwargs.items():
            setattr(self, key, value)
    def __repr__(self):
        attr = ', '.join(f'{key}={value}' for key, value in self.__dict__.
items() if key not in ('name', 'x', 'y'))
        if attr: return f"{self.name}({self.x}, {self.y}, {attr})"
        else: return f"{self.name}({self.x}, {self.y})"
    def __eq__(self, other):
        return all(getattr(self, key) == getattr(other, key, None) for key
in self.__dict__.keys())
    def __hash__(self):
        return hash(tuple(sorted(self.__dict__.items())))
class Agent(Entity): pass
class Key(Entity): pass
class Door(Entity): pass
class Goal(Entity): pass
class Wall(Entity): pass
class Box(Entity): pass
class Ball(Entity): pass
class Lava(Entity): pass
def get_entities_by_name(entities, name):
    return [ entity for entity in entities if entity.name == name ]
def get_entities_by_position(entities, x, y):
    return [ entity for entity in entities if entity.x == x and entity.y ==
y ]
def transition(state, action):
    """
    Args:
        state: the state of the environment
        action: the action to be executed
```

```
    Returns:
        next_state: the next state of the environment
    """
    raise NotImplementedError

‘‘‘
```

Please implement code to model the logic of the world as demonstrated by the
experiences. Please implement the code following the template. Feel free to
implement the helper functions you need. You can also implement the logic
for difference actions in different helper functions. However, you must
implement the ‘ transition ‘ function as the main function to be called by
the environment. The code needs to be directly runnable on the inputs as (
state, action) and return the next state in python as provided in the
experiences. Let's think step by step.

## F.2    Initializing the reward function

It asks LLMs to generate a reward function $(s, a, s') \to (r, d)$ for the mission $c$ following the code
template to model seven uniformly randomly sampled experience data $(s, a, s', r, d)$ in the replay
buffer for the mission $c$.

```
You are a robot exploring in an object-centric environment. Your goal is to
model the logic of the world in python. You will be provided experiences in
the format of (state, action, next_state, reward, done) tuples. You will
also be provided with a short natural language description that briefly
summarizes the difference between the state and the next state for each (
state, next_state) pair. You need to implement the python code to model the
logic of the world, as seen in the provided experiences. Please follow the
template to implement the code. The code needs to be directly runnable on
the (state, action, next_state) tuple and return the (reward, done) tuple in
 python as provided in the experiences.

You need to implement python code to model the logic of the world as seen in
 the following experiences for mission "use the key to open the door and
then get to the goal":

The action "turn left" transforms the state from
‘‘‘
Wall(0, 0) ;    Wall(1, 0) ;    Wall(2, 0) ;    Wall(3, 0) ;    Wall(4, 0) ;
Wall(0, 1) ;    empty ;         Wall(2, 1) ;    empty ;         Wall(4, 1) ;
Wall(0, 2) ;    Agent(1, 2, direction=(0, -1), carrying=None) ; Door(2, 2,
color=yellow, state=locked) ;         empty ;         Wall(4, 2) ;
Wall(0, 3) ;    Key(1, 3, color=yellow) ;       Wall(2, 3) ;    Goal(3, 3) ;
    Wall(4, 3) ;
Wall(0, 4) ;    Wall(1, 4) ;    Wall(2, 4) ;    Wall(3, 4) ;    Wall(4, 4) ;
‘‘‘
to
‘‘‘
Wall(0, 0) ;    Wall(1, 0) ;    Wall(2, 0) ;    Wall(3, 0) ;    Wall(4, 0) ;
Wall(0, 1) ;    empty ;         Wall(2, 1) ;    empty ;         Wall(4, 1) ;
Wall(0, 2) ;    Agent(1, 2, direction=(-1, 0), carrying=None) ; Door(2, 2,
color=yellow, state=locked) ;         empty ;         Wall(4, 2) ;
Wall(0, 3) ;    Key(1, 3, color=yellow) ;       Wall(2, 3) ;    Goal(3, 3) ;
    Wall(4, 3) ;
Wall(0, 4) ;    Wall(1, 4) ;    Wall(2, 4) ;    Wall(3, 4) ;    Wall(4, 4) ;
‘‘‘
The difference is
"""
```

The agent (direction=(0, -1)) at pos (1, 2) becomes an agent (direction=(-1, 0)).
"""
, the returned reward is ` 0.0 ` and the returned done is ` False `

The action "toggle" transforms the state from
```
Wall(0, 0) ;    Wall(1, 0) ;    Wall(2, 0) ;    Wall(3, 0) ;    Wall(4, 0) ;
Wall(0, 1) ;    empty ;         Wall(2, 1) ;    empty ;         Wall(4, 1) ;
Wall(0, 2) ;    Agent(1, 2, direction=(0, -1), carrying=None) ; Door(2, 2,
color=yellow, state=locked) ;          empty ;         Wall(4, 2) ;
Wall(0, 3) ;    Key(1, 3, color=yellow) ;       Wall(2, 3) ;    Goal(3, 3) ;
    Wall(4, 3) ;
Wall(0, 4) ;    Wall(1, 4) ;    Wall(2, 4) ;    Wall(3, 4) ;    Wall(4, 4) ;
```
to
```
Wall(0, 0) ;    Wall(1, 0) ;    Wall(2, 0) ;    Wall(3, 0) ;    Wall(4, 0) ;
Wall(0, 1) ;    empty ;         Wall(2, 1) ;    empty ;         Wall(4, 1) ;
Wall(0, 2) ;    Agent(1, 2, direction=(0, -1), carrying=None) ; Door(2, 2,
color=yellow, state=locked) ;          empty ;         Wall(4, 2) ;
Wall(0, 3) ;    Key(1, 3, color=yellow) ;       Wall(2, 3) ;    Goal(3, 3) ;
    Wall(4, 3) ;
Wall(0, 4) ;    Wall(1, 4) ;    Wall(2, 4) ;    Wall(3, 4) ;    Wall(4, 4) ;
```
The difference is
"""
Nothing happened
"""
, the returned reward is ` 0.0 ` and the returned done is ` False `

The action "turn right" transforms the state from
```
Wall(0, 0) ;    Wall(1, 0) ;    Wall(2, 0) ;    Wall(3, 0) ;    Wall(4, 0) ;
Wall(0, 1) ;    empty ;         Wall(2, 1) ;    empty ;         Wall(4, 1) ;
Wall(0, 2) ;    Agent(1, 2, direction=(1, 0), carrying=None) ;  Door(2, 2,
color=yellow, state=locked) ;          empty ;         Wall(4, 2) ;
Wall(0, 3) ;    Key(1, 3, color=yellow) ;       Wall(2, 3) ;    Goal(3, 3) ;
    Wall(4, 3) ;
Wall(0, 4) ;    Wall(1, 4) ;    Wall(2, 4) ;    Wall(3, 4) ;    Wall(4, 4) ;
```
to
```
Wall(0, 0) ;    Wall(1, 0) ;    Wall(2, 0) ;    Wall(3, 0) ;    Wall(4, 0) ;
Wall(0, 1) ;    empty ;         Wall(2, 1) ;    empty ;         Wall(4, 1) ;
Wall(0, 2) ;    Agent(1, 2, direction=(0, 1), carrying=None) ;  Door(2, 2,
color=yellow, state=locked) ;          empty ;         Wall(4, 2) ;
Wall(0, 3) ;    Key(1, 3, color=yellow) ;       Wall(2, 3) ;    Goal(3, 3) ;
    Wall(4, 3) ;
Wall(0, 4) ;    Wall(1, 4) ;    Wall(2, 4) ;    Wall(3, 4) ;    Wall(4, 4) ;
```
The difference is
"""
The agent (direction=(1, 0)) at pos (1, 2) becomes an agent (direction=(0, 1)).
"""
, the returned reward is ` 0.0 ` and the returned done is ` False `

The action "nothing" transforms the state from
```
Wall(0, 0) ;    Wall(1, 0) ;    Wall(2, 0) ;    Wall(3, 0) ;    Wall(4, 0) ;
Wall(0, 1) ;    empty ;         Wall(2, 1) ;    empty ;         Wall(4, 1) ;
```

```
Wall(0, 2) ;      Agent(1, 2, direction=(0, -1), carrying=None) ; Door(2, 2,
color=yellow, state=locked) ;         empty ;        Wall(4, 2) ;
Wall(0, 3) ;     Key(1, 3, color=yellow) ;        Wall(2, 3) ;     Goal(3, 3) ;
    Wall(4, 3) ;
Wall(0, 4) ;     Wall(1, 4) ;     Wall(2, 4) ;     Wall(3, 4) ;     Wall(4, 4) ;
```
to
```
Wall(0, 0) ;     Wall(1, 0) ;     Wall(2, 0) ;     Wall(3, 0) ;     Wall(4, 0) ;
Wall(0, 1) ;     empty ;          Wall(2, 1) ;     empty ;          Wall(4, 1) ;
Wall(0, 2) ;      Agent(1, 2, direction=(0, -1), carrying=None) ; Door(2, 2,
color=yellow, state=locked) ;         empty ;        Wall(4, 2) ;
Wall(0, 3) ;     Key(1, 3, color=yellow) ;        Wall(2, 3) ;     Goal(3, 3) ;
    Wall(4, 3) ;
Wall(0, 4) ;     Wall(1, 4) ;     Wall(2, 4) ;     Wall(3, 4) ;     Wall(4, 4) ;
```
The difference is
"""
Nothing happened
"""
, the returned reward is ' 0.0 ' and the returned done is ' False '

The action "drop" transforms the state from
```
Wall(0, 0) ;     Wall(1, 0) ;     Wall(2, 0) ;     Wall(3, 0) ;     Wall(4, 0) ;
Wall(0, 1) ;     empty ;          Wall(2, 1) ;     empty ;          Wall(4, 1) ;
Wall(0, 2) ;      Agent(1, 2, direction=(0, -1), carrying=None) ; Door(2, 2,
color=yellow, state=locked) ;         empty ;        Wall(4, 2) ;
Wall(0, 3) ;     Key(1, 3, color=yellow) ;        Wall(2, 3) ;     Goal(3, 3) ;
    Wall(4, 3) ;
Wall(0, 4) ;     Wall(1, 4) ;     Wall(2, 4) ;     Wall(3, 4) ;     Wall(4, 4) ;
```
to
```
Wall(0, 0) ;     Wall(1, 0) ;     Wall(2, 0) ;     Wall(3, 0) ;     Wall(4, 0) ;
Wall(0, 1) ;     empty ;          Wall(2, 1) ;     empty ;          Wall(4, 1) ;
Wall(0, 2) ;      Agent(1, 2, direction=(0, -1), carrying=None) ; Door(2, 2,
color=yellow, state=locked) ;         empty ;        Wall(4, 2) ;
Wall(0, 3) ;     Key(1, 3, color=yellow) ;        Wall(2, 3) ;     Goal(3, 3) ;
    Wall(4, 3) ;
Wall(0, 4) ;     Wall(1, 4) ;     Wall(2, 4) ;     Wall(3, 4) ;     Wall(4, 4) ;
```
The difference is
"""
Nothing happened
"""
, the returned reward is ' 0.0 ' and the returned done is ' False '

The action "turn left" transforms the state from
```
Wall(0, 0) ;     Wall(1, 0) ;     Wall(2, 0) ;     Wall(3, 0) ;     Wall(4, 0) ;
Wall(0, 1) ;     empty ;          Wall(2, 1) ;     empty ;          Wall(4, 1) ;
Wall(0, 2) ;      Agent(1, 2, direction=(-1, 0), carrying=None) ; Door(2, 2,
color=yellow, state=locked) ;         empty ;        Wall(4, 2) ;
Wall(0, 3) ;     Key(1, 3, color=yellow) ;        Wall(2, 3) ;     Goal(3, 3) ;
    Wall(4, 3) ;
Wall(0, 4) ;     Wall(1, 4) ;     Wall(2, 4) ;     Wall(3, 4) ;     Wall(4, 4) ;
```
to
```
Wall(0, 0) ;     Wall(1, 0) ;     Wall(2, 0) ;     Wall(3, 0) ;     Wall(4, 0) ;
Wall(0, 1) ;     empty ;          Wall(2, 1) ;     empty ;          Wall(4, 1) ;
```

```
Wall(0, 2) ;    Agent(1, 2, direction=(0, 1), carrying=None) ;  Door(2, 2,
color=yellow, state=locked) ;         empty ;        Wall(4, 2) ;
Wall(0, 3) ;    Key(1, 3, color=yellow) ;       Wall(2, 3) ;    Goal(3, 3) ;
    Wall(4, 3) ;
Wall(0, 4) ;    Wall(1, 4) ;    Wall(2, 4) ;    Wall(3, 4) ;    Wall(4, 4) ;
‘‘‘
The difference is
"""
The agent (direction=(-1, 0)) at pos (1, 2) becomes an agent (direction=(0,
1)).
"""
, the returned reward is ‘ 0.0 ‘ and the returned done is ‘ False ‘

The action "turn right" transforms the state from
‘‘‘
Wall(0, 0) ;    Wall(1, 0) ;    Wall(2, 0) ;    Wall(3, 0) ;    Wall(4, 0) ;
Wall(0, 1) ;    empty ;         Wall(2, 1) ;    empty ;         Wall(4, 1) ;
Wall(0, 2) ;    Agent(1, 2, direction=(-1, 0), carrying=None) ; Door(2, 2,
color=yellow, state=locked) ;         empty ;        Wall(4, 2) ;
Wall(0, 3) ;    Key(1, 3, color=yellow) ;       Wall(2, 3) ;    Goal(3, 3) ;
    Wall(4, 3) ;
Wall(0, 4) ;    Wall(1, 4) ;    Wall(2, 4) ;    Wall(3, 4) ;    Wall(4, 4) ;
‘‘‘
to
‘‘‘
Wall(0, 0) ;    Wall(1, 0) ;    Wall(2, 0) ;    Wall(3, 0) ;    Wall(4, 0) ;
Wall(0, 1) ;    empty ;         Wall(2, 1) ;    empty ;         Wall(4, 1) ;
Wall(0, 2) ;    Agent(1, 2, direction=(0, -1), carrying=None) ; Door(2, 2,
color=yellow, state=locked) ;         empty ;        Wall(4, 2) ;
Wall(0, 3) ;    Key(1, 3, color=yellow) ;       Wall(2, 3) ;    Goal(3, 3) ;
    Wall(4, 3) ;
Wall(0, 4) ;    Wall(1, 4) ;    Wall(2, 4) ;    Wall(3, 4) ;    Wall(4, 4) ;
‘‘‘
The difference is
"""
The agent (direction=(-1, 0)) at pos (1, 2) becomes an agent (direction=(0,
-1)).
"""
, the returned reward is ‘ 0.0 ‘ and the returned done is ‘ False ‘

Please implement code to model the logic of the world as demonstrated by the
 experiences. Here is the template for the reward function. Please implement
 the reward function following the template. The code needs to be directly
runnable on the inputs of (state, action, next_state) and return (reward,
done) in python as provided in the experiences.

‘‘‘

class Entity:
    def __init__(self, x, y, **kwargs):
        self.name = self.__class__.__name__
        self.x = x
        self.y = y
        for key, value in kwargs.items():
            setattr(self, key, value)
    def __repr__(self):
        attr = ', '.join(f'{key}={value}' for key, value in self.__dict__.
items() if key not in ('name', 'x', 'y'))
        if attr: return f"{self.name}({self.x}, {self.y}, {attr})"
        else: return f"{self.name}({self.x}, {self.y})"
    def __eq__(self, other):
        return all(getattr(self, key) == getattr(other, key, None) for key
in self.__dict__.keys())
```

```python
    def __hash__(self):
        return hash(tuple(sorted(self.__dict__.items())))
class Agent(Entity): pass
class Key(Entity): pass
class Door(Entity): pass
class Goal(Entity): pass
class Wall(Entity): pass
class Box(Entity): pass
class Ball(Entity): pass
class Lava(Entity): pass
def get_entities_by_name(entities, name):
    return [ entity for entity in entities if entity.name == name ]
def get_entities_by_position(entities, x, y):
    return [ entity for entity in entities if entity.x == x and entity.y ==
y ]
def reward_func(state, action, next_state):
    """
    Args:
        state: the state of the environment
        action: the action to be executed
        next_state: the next state of the environment
    Returns:
        reward: the reward of the action
        done: whether the episode is done
    """
    raise NotImplementedError

‘‘‘

Please implement code to model the logic of the world as demonstrated by the
experiences. Please implement the code following the template. You must
implement the ‘ reward_func ‘ function as the main function to be called by
the environment. The code needs to be directly runnable on the inputs as (
state, action, next_state) and return (reward, done) in python as provided
in the experiences. Let's think step by step.
```

## F.3   Refining the transition function

It asks LLMs to refine a partially correct transition function by providing it with a data point that it fails to model as well as a few other data points that it succeeds in modelling. We also provide the wrong prediction by the partially correct code or the error message during execution.

```python
You are a robot exploring in an object-centric environment. Your goal is to
model the logic of the world in python. You have tried it before and came up
 with one partially correct solution. However, it is not perfect. They can
model the logic for some experiences but failed for others. You need to
improve your code to model the logic of the world for all the experiences.
The new code needs to be directly runnable on the (state, action) pair and
return the next state in python as provided in the experiences.

Here is the partially correct solution you came up with. It can model the
logic for some experiences but failed for others. You need to improve your
code to model the logic of the world for all the experiences. The new code
needs to be directly runnable on the (state, action) pair and return the
next state in python as provided in the experiences.

‘‘‘

class Entity:
    def __init__(self, x, y, **kwargs):
```

```python
        self.name = self.__class__.__name__
        self.x = x
        self.y = y
        for key, value in kwargs.items():
            setattr(self, key, value)
    def __repr__(self):
        attr = ', '.join(f'{key}={value}' for key, value in self.__dict__.
items() if key not in ('name', 'x', 'y'))
        if attr: return f"{self.name}({self.x}, {self.y}, {attr})"
        else: return f"{self.name}({self.x}, {self.y})"
    def __eq__(self, other):
        return all(getattr(self, key) == getattr(other, key, None) for key
in self.__dict__.keys())
    def __hash__(self):
        return hash(tuple(sorted(self.__dict__.items())))
class Agent(Entity): pass
class Key(Entity): pass
class Door(Entity): pass
class Goal(Entity): pass
class Wall(Entity): pass
class Box(Entity): pass
class Ball(Entity): pass
class Lava(Entity): pass
import copy
def get_entities_by_name(entities, name):
    return [ entity for entity in entities if entity.name == name ]
def get_entities_by_position(entities, x, y):
    return [ entity for entity in entities if entity.x == x and entity.y ==
y ]
def transition(state, action):
    next_state = copy.deepcopy(state)
    agent = get_entities_by_name(next_state, 'Agent')[0]
    # Determine agent's next position based on action
    if action == 'move forward':
        next_pos = (agent.x + agent.direction[0], agent.y + agent.direction
[1])
        # Check if the next position isn't a wall
        if not any(isinstance(entity, Wall) for entity in
get_entities_by_position(state, *next_pos)):
            # If agent is in front of a door and has the right color key,
unlock the door
            if any(isinstance(entity, Door) and entity.color == agent.
carrying.color for entity in get_entities_by_position(state, *next_pos)):
                if action == 'toggle':
                    agent.carrying = None  # Drop the key
            else:
                agent.x, agent.y = next_pos  # Move forward
    elif action == 'pick up':
        # Pick up a key if there is a key at the agent's position
        for entity in get_entities_by_position(next_state, agent.x, agent.y)
:
            if isinstance(entity, Key):
                agent.carrying = entity
                next_state.remove(entity)
                break
    elif action == 'drop':
        # Drop the key at the agent's position if the agent is carrying a
key
        if agent.carrying is not None:
            dropped_key = Key(agent.x, agent.y, color=agent.carrying.color)
            next_state.append(dropped_key)
            agent.carrying = None
    elif action in ['turn left', 'turn right']:
```

```
            # Existing code for turn left/right here
            pass
    elif action == 'toggle':
            # Existing code for toggle here
            pass
    return next_state
```

```

```

The given code cannot model the logic of the world for all the experiences.
Here are some experiences that the code have successfully
modeled.

The action "toggle" transforms the state from
```
Wall(0, 0) ;    Wall(1, 0) ;    Wall(2, 0) ;    Wall(3, 0) ;    Wall(4, 0) ;
Wall(0, 1) ;    empty ;         Wall(2, 1) ;    empty ;         Wall(4, 1) ;
Wall(0, 2) ;    Agent(1, 2, direction=(0, -1), carrying=None) ; Door(2, 2,
color=yellow, state=locked) ;        empty ;         Wall(4, 2) ;
Wall(0, 3) ;    Key(1, 3, color=yellow) ;       Wall(2, 3) ;    Goal(3, 3) ;
    Wall(4, 3) ;
Wall(0, 4) ;    Wall(1, 4) ;    Wall(2, 4) ;    Wall(3, 4) ;    Wall(4, 4) ;
```
to
```
Wall(0, 0) ;    Wall(1, 0) ;    Wall(2, 0) ;    Wall(3, 0) ;    Wall(4, 0) ;
Wall(0, 1) ;    empty ;         Wall(2, 1) ;    empty ;         Wall(4, 1) ;
Wall(0, 2) ;    Agent(1, 2, direction=(0, -1), carrying=None) ; Door(2, 2,
color=yellow, state=locked) ;        empty ;         Wall(4, 2) ;
Wall(0, 3) ;    Key(1, 3, color=yellow) ;       Wall(2, 3) ;    Goal(3, 3) ;
    Wall(4, 3) ;
Wall(0, 4) ;    Wall(1, 4) ;    Wall(2, 4) ;    Wall(3, 4) ;    Wall(4, 4) ;
```
The difference is
"""
Nothing happened
"""

The action "drop" transforms the state from
```
Wall(0, 0) ;    Wall(1, 0) ;    Wall(2, 0) ;    Wall(3, 0) ;    Wall(4, 0) ;
Wall(0, 1) ;    empty ;         Wall(2, 1) ;    empty ;         Wall(4, 1) ;
Wall(0, 2) ;    Agent(1, 2, direction=(0, -1), carrying=None) ; Door(2, 2,
color=yellow, state=locked) ;        empty ;         Wall(4, 2) ;
Wall(0, 3) ;    Key(1, 3, color=yellow) ;       Wall(2, 3) ;    Goal(3, 3) ;
    Wall(4, 3) ;
Wall(0, 4) ;    Wall(1, 4) ;    Wall(2, 4) ;    Wall(3, 4) ;    Wall(4, 4) ;
```
to
```
Wall(0, 0) ;    Wall(1, 0) ;    Wall(2, 0) ;    Wall(3, 0) ;    Wall(4, 0) ;
Wall(0, 1) ;    empty ;         Wall(2, 1) ;    empty ;         Wall(4, 1) ;
Wall(0, 2) ;    Agent(1, 2, direction=(0, -1), carrying=None) ; Door(2, 2,
color=yellow, state=locked) ;        empty ;         Wall(4, 2) ;
Wall(0, 3) ;    Key(1, 3, color=yellow) ;       Wall(2, 3) ;    Goal(3, 3) ;
    Wall(4, 3) ;
Wall(0, 4) ;    Wall(1, 4) ;    Wall(2, 4) ;    Wall(3, 4) ;    Wall(4, 4) ;
```
The difference is
"""
Nothing happened
"""

```
The action "nothing" transforms the state from
```
```
Wall(0, 0) ;    Wall(1, 0) ;    Wall(2, 0) ;    Wall(3, 0) ;    Wall(4, 0) ;
Wall(0, 1) ;    empty ;         Wall(2, 1) ;    empty ;         Wall(4, 1) ;
Wall(0, 2) ;    Agent(1, 2, direction=(0, -1), carrying=None) ; Door(2, 2,
color=yellow, state=locked) ;        empty ;        Wall(4, 2) ;
Wall(0, 3) ;    Key(1, 3, color=yellow) ;       Wall(2, 3) ;    Goal(3, 3) ;
    Wall(4, 3) ;
Wall(0, 4) ;    Wall(1, 4) ;    Wall(2, 4) ;    Wall(3, 4) ;    Wall(4, 4) ;
```
```
to
```
```
Wall(0, 0) ;    Wall(1, 0) ;    Wall(2, 0) ;    Wall(3, 0) ;    Wall(4, 0) ;
Wall(0, 1) ;    empty ;         Wall(2, 1) ;    empty ;         Wall(4, 1) ;
Wall(0, 2) ;    Agent(1, 2, direction=(0, -1), carrying=None) ; Door(2, 2,
color=yellow, state=locked) ;        empty ;        Wall(4, 2) ;
Wall(0, 3) ;    Key(1, 3, color=yellow) ;       Wall(2, 3) ;    Goal(3, 3) ;
    Wall(4, 3) ;
Wall(0, 4) ;    Wall(1, 4) ;    Wall(2, 4) ;    Wall(3, 4) ;    Wall(4, 4) ;
```
```
The difference is
"""
Nothing happened
"""
```

```
The action "turn left" should transform the state from
```
```
Wall(0, 0) ;    Wall(1, 0) ;    Wall(2, 0) ;    Wall(3, 0) ;    Wall(4, 0) ;
Wall(0, 1) ;    empty ;         Wall(2, 1) ;    empty ;         Wall(4, 1) ;
Wall(0, 2) ;    Agent(1, 2, direction=(0, 1), carrying=None) ;  Door(2, 2,
color=yellow, state=locked) ;        empty ;        Wall(4, 2) ;
Wall(0, 3) ;    Key(1, 3, color=yellow) ;       Wall(2, 3) ;    Goal(3, 3) ;
    Wall(4, 3) ;
Wall(0, 4) ;    Wall(1, 4) ;    Wall(2, 4) ;    Wall(3, 4) ;    Wall(4, 4) ;
```
```
to
```
```
Wall(0, 0) ;    Wall(1, 0) ;    Wall(2, 0) ;    Wall(3, 0) ;    Wall(4, 0) ;
Wall(0, 1) ;    empty ;         Wall(2, 1) ;    empty ;         Wall(4, 1) ;
Wall(0, 2) ;    Agent(1, 2, direction=(1, 0), carrying=None) ;  Door(2, 2,
color=yellow, state=locked) ;        empty ;        Wall(4, 2) ;
Wall(0, 3) ;    Key(1, 3, color=yellow) ;       Wall(2, 3) ;    Goal(3, 3) ;
    Wall(4, 3) ;
Wall(0, 4) ;    Wall(1, 4) ;    Wall(2, 4) ;    Wall(3, 4) ;    Wall(4, 4) ;
```
```
The difference is
"""
The agent (direction=(0, 1)) at pos (1, 2) becomes an agent (direction=(1,
0)).
"""
```
```
However, the implementation is wrong because it returns state as
```
```
Wall(0, 0) ;    Wall(1, 0) ;    Wall(2, 0) ;    Wall(3, 0) ;    Wall(4, 0) ;
Wall(0, 1) ;    empty ;         Wall(2, 1) ;    empty ;         Wall(4, 1) ;
Wall(0, 2) ;    Agent(1, 2, direction=(0, 1), carrying=None) ;  Door(2, 2,
color=yellow, state=locked) ;        empty ;        Wall(4, 2) ;
Wall(0, 3) ;    Key(1, 3, color=yellow) ;       Wall(2, 3) ;    Goal(3, 3) ;
    Wall(4, 3) ;
Wall(0, 4) ;    Wall(1, 4) ;    Wall(2, 4) ;    Wall(3, 4) ;    Wall(4, 4) ;
```

```
For this failed experience, do you know what is different between the true
transitions from the environment and the predictions from the code? Do you
know why the environment behaves in this way? Do you know why the code
behaves differently from the environment? Which part of the code causes the
problem? How to fix it? Please improve your code to model the logic of the
world for all the experiences,
accordingly. Please implement the code following the template. Feel free to
implement any helper functions you need. You can also implement the logic
for difference actions in different helper functions. However, you must
implement the ' transition ' function as the main function to be called by
the environment. The code needs to be directly runnable on the (state,
action) tuple and return the new state in python as provided in the
experiences. If the code is too long, try to refactor it to be shorter.
```

## F.4 Refine the reward function

It asks LLMs to refine a partially correct reward function by providing it with a data point that it fails to model as well as a few other data points that it succeeds in modelling. We also provide the wrong prediction by the partially correct code or the error message during execution.

```
You are a robot exploring in an object-centric environment. Your goal is to
model the logic of the world in python. You have tried it before and came up
 with one partially correct solution. However, it is not perfect. They can
model the logic for some experiences but failed for others. You need to
improve your code to model the logic of the world for all the experiences.
The new code needs to be directly runnable on the (state, action, next_state
) tuple and return the (reward, done) tuple in python as provided in the
experiences.

Here is the partially correct solution you came up with for mission "use the
 key to open the door and then get to the goal". It can model the logic for
some experiences but failed for others. You need to improve your code to
model the logic of the world for all the experiences. The new code need to
be directly runnable on the (state, action, next_state) tuple and return the
 (reward, done) tuple in python as provided in the experiences.

' ' '

class Entity:
    def __init__(self, x, y, **kwargs):
        self.name = self.__class__.__name__
        self.x = x
        self.y = y
        for key, value in kwargs.items():
            setattr(self, key, value)
    def __repr__(self):
        attr = ', '.join(f'{key}={value}' for key, value in self.__dict__.
items() if key not in ('name', 'x', 'y'))
        if attr: return f"{self.name}({self.x}, {self.y}, {attr})"
        else: return f"{self.name}({self.x}, {self.y})"
    def __eq__(self, other):
        return all(getattr(self, key) == getattr(other, key, None) for key
in self.__dict__.keys())
    def __hash__(self):
        return hash(tuple(sorted(self.__dict__.items())))
class Agent(Entity): pass
class Key(Entity): pass
class Door(Entity): pass
class Goal(Entity): pass
class Wall(Entity): pass
```

```python
class Box(Entity): pass
class Ball(Entity): pass
class Lava(Entity): pass
def get_entities_by_position(entities, x, y):
    return [ entity for entity in entities if entity.x == x and entity.y ==
y ]
def reward_func(state, action, next_state):
    state_set = set(state)
    next_state_set = set(next_state)
    agent = [e for e in state_set if isinstance(e, Agent)][0]
    agent_next = [e for e in next_state_set if isinstance(e, Agent)][0]
    on_goal = any(isinstance(entity, Goal) for entity in
get_entities_by_position(next_state, agent_next.x, agent_next.y))
    done = on_goal
    if state_set == next_state_set:
        reward = -0.1  # Small negative reward for no-op actions to
encourage faster solution
    elif done:
        reward = 1.0  # Reward for reaching the goal
    else:
        reward = 0.0  # No reward in other cases
    return reward, done
```

‘‘‘

The given code cannot model the logic of the world for all the experiences.
Here are some experiences that the code has successfully
modeled.

The action "turn right" transforms the state from
‘‘‘
```
Wall(0, 0) ;    Wall(1, 0) ;    Wall(2, 0) ;    Wall(3, 0) ;    Wall(4, 0) ;
Wall(0, 1) ;    empty ;         Wall(2, 1) ;    empty ;         Wall(4, 1) ;
Wall(0, 2) ;    Agent(1, 2, direction=(-1, 0), carrying=None) ; Door(2, 2,
color=yellow, state=locked) ;          empty ;          Wall(4, 2) ;
Wall(0, 3) ;    Key(1, 3, color=yellow) ;      Wall(2, 3) ;    Goal(3, 3) ;
    Wall(4, 3) ;
Wall(0, 4) ;    Wall(1, 4) ;    Wall(2, 4) ;    Wall(3, 4) ;    Wall(4, 4) ;
```
‘‘‘
to
‘‘‘
```
Wall(0, 0) ;    Wall(1, 0) ;    Wall(2, 0) ;    Wall(3, 0) ;    Wall(4, 0) ;
Wall(0, 1) ;    empty ;         Wall(2, 1) ;    empty ;         Wall(4, 1) ;
Wall(0, 2) ;    Agent(1, 2, direction=(0, -1), carrying=None) ; Door(2, 2,
color=yellow, state=locked) ;          empty ;          Wall(4, 2) ;
Wall(0, 3) ;    Key(1, 3, color=yellow) ;      Wall(2, 3) ;    Goal(3, 3) ;
    Wall(4, 3) ;
Wall(0, 4) ;    Wall(1, 4) ;    Wall(2, 4) ;    Wall(3, 4) ;    Wall(4, 4) ;
```
‘‘‘
The difference is
"""
The agent (direction=(-1, 0)) at pos (1, 2) becomes an agent (direction=(0,
-1)).
"""
, the returned reward is ‘ 0.0 ‘ and the returned done is ‘ False ‘

The action "turn left" transforms the state from
‘‘‘
```
Wall(0, 0) ;    Wall(1, 0) ;    Wall(2, 0) ;    Wall(3, 0) ;    Wall(4, 0) ;
Wall(0, 1) ;    empty ;         Wall(2, 1) ;    empty ;         Wall(4, 1) ;
Wall(0, 2) ;    Agent(1, 2, direction=(-1, 0), carrying=None) ; Door(2, 2,
color=yellow, state=locked) ;          empty ;          Wall(4, 2) ;
```

```
Wall(0, 3) ;     Key(1, 3, color=yellow) ;        Wall(2, 3) ;     Goal(3, 3) ;
    Wall(4, 3) ;
Wall(0, 4) ;     Wall(1, 4) ;     Wall(2, 4) ;     Wall(3, 4) ;     Wall(4, 4) ;
‘‘‘
to
‘‘‘
Wall(0, 0) ;     Wall(1, 0) ;     Wall(2, 0) ;     Wall(3, 0) ;     Wall(4, 0) ;
Wall(0, 1) ;     empty ;          Wall(2, 1) ;     empty ;          Wall(4, 1) ;
Wall(0, 2) ;     Agent(1, 2, direction=(0, 1), carrying=None) ;  Door(2, 2,
color=yellow, state=locked) ;          empty ;        Wall(4, 2) ;
Wall(0, 3) ;     Key(1, 3, color=yellow) ;        Wall(2, 3) ;     Goal(3, 3) ;
    Wall(4, 3) ;
Wall(0, 4) ;     Wall(1, 4) ;     Wall(2, 4) ;     Wall(3, 4) ;     Wall(4, 4) ;
‘‘‘
The difference is
"""
The agent (direction=(-1, 0)) at pos (1, 2) becomes an agent (direction=(0,
1)).
"""
, the returned reward is ‘ 0.0 ‘ and the returned done is ‘ False ‘

The action "turn left" transforms the state from
‘‘‘
Wall(0, 0) ;     Wall(1, 0) ;     Wall(2, 0) ;     Wall(3, 0) ;     Wall(4, 0) ;
Wall(0, 1) ;     empty ;          Wall(2, 1) ;     empty ;          Wall(4, 1) ;
Wall(0, 2) ;     Agent(1, 2, direction=(0, -1), carrying=None) ; Door(2, 2,
color=yellow, state=locked) ;          empty ;        Wall(4, 2) ;
Wall(0, 3) ;     Key(1, 3, color=yellow) ;        Wall(2, 3) ;     Goal(3, 3) ;
    Wall(4, 3) ;
Wall(0, 4) ;     Wall(1, 4) ;     Wall(2, 4) ;     Wall(3, 4) ;     Wall(4, 4) ;
‘‘‘
to
‘‘‘
Wall(0, 0) ;     Wall(1, 0) ;     Wall(2, 0) ;     Wall(3, 0) ;     Wall(4, 0) ;
Wall(0, 1) ;     empty ;          Wall(2, 1) ;     empty ;          Wall(4, 1) ;
Wall(0, 2) ;     Agent(1, 2, direction=(-1, 0), carrying=None) ; Door(2, 2,
color=yellow, state=locked) ;          empty ;        Wall(4, 2) ;
Wall(0, 3) ;     Key(1, 3, color=yellow) ;        Wall(2, 3) ;     Goal(3, 3) ;
    Wall(4, 3) ;
Wall(0, 4) ;     Wall(1, 4) ;     Wall(2, 4) ;     Wall(3, 4) ;     Wall(4, 4) ;
‘‘‘
The difference is
"""
The agent (direction=(0, -1)) at pos (1, 2) becomes an agent (direction=(-1,
 0)).
"""
, the returned reward is ‘ 0.0 ‘ and the returned done is ‘ False ‘
```

==Here is an example of experiences that the code failed to model.==

```
The action "toggle" should transform the state from
‘‘‘
Wall(0, 0) ;     Wall(1, 0) ;     Wall(2, 0) ;     Wall(3, 0) ;     Wall(4, 0) ;
Wall(0, 1) ;     empty ;          Wall(2, 1) ;     empty ;          Wall(4, 1) ;
Wall(0, 2) ;     Agent(1, 2, direction=(0, 1), carrying=None) ;  Door(2, 2,
color=yellow, state=locked) ;          empty ;        Wall(4, 2) ;
Wall(0, 3) ;     Key(1, 3, color=yellow) ;        Wall(2, 3) ;     Goal(3, 3) ;
    Wall(4, 3) ;
Wall(0, 4) ;     Wall(1, 4) ;     Wall(2, 4) ;     Wall(3, 4) ;     Wall(4, 4) ;
‘‘‘
to
‘‘‘
Wall(0, 0) ;     Wall(1, 0) ;     Wall(2, 0) ;     Wall(3, 0) ;     Wall(4, 0) ;
```

```
Wall(0, 1) ;     empty ;          Wall(2, 1) ;     empty ;         Wall(4, 1) ;
Wall(0, 2) ;     Agent(1, 2, direction=(0, 1), carrying=None) ;  Door(2, 2,
color=yellow, state=locked) ;          empty ;        Wall(4, 2) ;
Wall(0, 3) ;    Key(1, 3, color=yellow) ;       Wall(2, 3) ;    Goal(3, 3) ;
    Wall(4, 3) ;
Wall(0, 4) ;    Wall(1, 4) ;    Wall(2, 4) ;    Wall(3, 4) ;    Wall(4, 4) ;
‘‘‘
The difference is
"""
Nothing happened
"""
, the returned reward should be ‘ 0.0 ‘ and the returned done should be ‘
False ‘.
However, the implementation is wrong because it returns the predicted reward
 as ‘ -0.1 ‘ instead of the correct reward as ‘ 0.0 ‘.

For this failed experience, do you know what is different between the true
rewards and dones from the environment and the predictions from the code? Do
 you know why the environment behaves in this way? Do you know why the code
behaves differently from the environment? Which part of the code causes the
problem? How to fix it? Please improve your code to model the logic of the
world for all the experiences,
accordingly. Please implement the code following the template. You must
implement the ‘ reward_func ‘ function as the main function to be called by
the environment. The code needs to be directly runnable on the (state,
action, next_state) tuple and return (reward, done) in python as provided in
 the experiences. If the code is too long, try to refactor it to be shorter.
```

## F.5  Generating reward functions for new goals

It asks LLMs to generate new reward functions for new goals, given sample code that are synthesized for previous goals.

```
You are a robot exploring in an object-centric environment. Your goal is to
model the logic of the world in python, specifically the reward function
that maps (state, action, next_state) to (reward, done). You will to be
given the mission for the environment you are going to act in, as well as a
few sample code from the other environments. You need to implement the new
reward function for the new environment you are going to act in. The new
code needs to be directly runnable on (state, action, next_state) and return
 (reward, done) in python.

Here is a few sample code for the reward function in other environments.
Please check them in detail and think about how to implement the reward
function for mission "pick up the yellow box" in the new environment. The
code needs to be directly runnable on (state, action, next_state) and return
 (reward, done) in python.

The reward function code for mission "pick up the grey box" is:
‘‘‘

state = [{"name":"Agent", "x":1, "y":1, "direction":(1,0), "carrying": {"
name":"Key", "x":None, "y":None, "color":"red"}},
        {"name":"Door", "x":2, "y":2, "color":"red", "state":"locked"},
        {"name":"Wall", "x":0, "y":0}]
action = "toggle"
next_state = [{"name":"Agent", "x":1, "y":1, "direction":(1,0), "carrying":
{"name":"Key", "x":None, "y":None, "color":"red"}},
            {"name":"Door", "x":2, "y":2, "color":"red", "state":"open"},
            {"name":"Wall", "x":0, "y":0}]
```

```python
class Entity:
    def __init__(self, x, y, **kwargs):
        self.name = self.__class__.__name__
        self.x = x
        self.y = y
        for key, value in kwargs.items():
            setattr(self, key, value)
    def __repr__(self):
        attr = ', '.join(f'{key}={value}' for key, value in self.__dict__.
items() if key not in ('name', 'x', 'y'))
        if attr: return f"{self.name}({self.x}, {self.y}, {attr})"
        else: return f"{self.name}({self.x}, {self.y})"
    def __eq__(self, other):
        return all(getattr(self, key) == getattr(other, key, None) for key
in self.__dict__.keys())
    def __hash__(self):
        return hash(tuple(sorted(self.__dict__.items())))
class Agent(Entity): pass
class Key(Entity): pass
class Door(Entity): pass
class Goal(Entity): pass
class Wall(Entity): pass
class Box(Entity): pass
class Ball(Entity): pass
class Lava(Entity): pass
def get_entities_by_name(entities, name):
    return [ entity for entity in entities if entity.name == name ]
def get_entities_by_position(entities, x, y):
    return [ entity for entity in entities if entity.x == x and entity.y ==
y ]
def reward_func(state, action, next_state):
    """
    Args:
        state: the state of the environment
        action: the action to be executed
        next_state: the next state of the environment
    Returns:
        reward: the reward of the action
        done: whether the episode is done
    """
    reward = 0.0  # initialise reward as 0.0 for all actions
    done = False # initialise done as False for all actions
    # extract the agent from the current and next state
    agent = get_entities_by_name(state, 'Agent')[0]
    next_agent = get_entities_by_name(next_state, 'Agent')[0]
    # If the agent picks up the grey box in the next state, the reward is
1.0 and the episode is done
    if next_agent.carrying and isinstance(next_agent.carrying, Box) and
next_agent.carrying.color == 'grey':
        reward = 1.0
        done = True
    return reward, done

‘‘‘

The reward function code for mission "pick up the purple box" is:
‘‘‘

state = [{"name":"Agent", "x":1, "y":1, "direction":(1,0), "carrying": {"
name":"Key", "x":None, "y":None, "color":"red"}},
        {"name":"Door", "x":2, "y":2, "color":"red", "state":"locked"},
        {"name":"Wall", "x":0, "y":0}]
action = "toggle"
```

```
next_state = [{"name":"Agent", "x":1, "y":1, "direction":(1,0), "carrying":
{"name":"Key", "x":None, "y":None, "color":"red"}},
              {"name":"Door", "x":2, "y":2, "color":"red", "state":"open"},
              {"name":"Wall", "x":0, "y":0}]
class Entity:
    def __init__(self, x, y, **kwargs):
        self.name = self.__class__.__name__
        self.x = x
        self.y = y
        for key, value in kwargs.items():
            setattr(self, key, value)
    def __repr__(self):
        attr = ', '.join(f'{key}={value}' for key, value in self.__dict__.
items() if key not in ('name', 'x', 'y'))
        if attr: return f"{self.name}({self.x}, {self.y}, {attr})"
        else: return f"{self.name}({self.x}, {self.y})"
    def __eq__(self, other):
        return all(getattr(self, key) == getattr(other, key, None) for key
in self.__dict__.keys())
    def __hash__(self):
        return hash(tuple(sorted(self.__dict__.items())))
class Agent(Entity): pass
class Key(Entity): pass
class Door(Entity): pass
class Goal(Entity): pass
class Wall(Entity): pass
class Box(Entity): pass
class Ball(Entity): pass
class Lava(Entity): pass
def get_entities_by_name(entities, name):
    return [ entity for entity in entities if entity.name == name ]
def get_entities_by_position(entities, x, y):
    return [ entity for entity in entities if entity.x == x and entity.y ==
y ]
def reward_func(state, action, next_state):
    """
    Args:
        state: the state of the environment
        action: the action to be executed
        next_state: the next state of the environment
    Returns:
        reward: the reward of the action
        done: whether the episode is done
    """
    reward = 0.0  # initialise reward as 0.0 for all actions
    done = False # initialise done as False for all actions
    # extract the agent from the current and next state
    agent = get_entities_by_name(state, 'Agent')[0]
    next_agent = get_entities_by_name(next_state, 'Agent')[0]
    # If the agent picks up the purple box in the next state, the reward is
1.0 and the episode is done
    if next_agent.carrying and isinstance(next_agent.carrying, Box) and
next_agent.carrying.color == 'purple':
        reward = 1.0
        done = True
    return reward, done

‘‘‘

The reward function code for mission "pick up the green box" is:
‘‘‘
```

```python
state = [{"name":"Agent", "x":1, "y":1, "direction":(1,0), "carrying": {"
name":"Key", "x":None, "y":None, "color":"red"}},
         {"name":"Door", "x":2, "y":2, "color":"red", "state":"locked"},
         {"name":"Wall", "x":0, "y":0}]
action = "toggle"
next_state = [{"name":"Agent", "x":1, "y":1, "direction":(1,0), "carrying":
{"name":"Key", "x":None, "y":None, "color":"red"}},
              {"name":"Door", "x":2, "y":2, "color":"red", "state":"open"},
              {"name":"Wall", "x":0, "y":0}]
class Entity:
    def __init__(self, x, y, **kwargs):
        self.name = self.__class__.__name__
        self.x = x
        self.y = y
        for key, value in kwargs.items():
            setattr(self, key, value)
    def __repr__(self):
        attr = ', '.join(f'{key}={value}' for key, value in self.__dict__.
items() if key not in ('name', 'x', 'y'))
        if attr: return f"{self.name}({self.x}, {self.y}, {attr})"
        else: return f"{self.name}({self.x}, {self.y})"
    def __eq__(self, other):
        return all(getattr(self, key) == getattr(other, key, None) for key
in self.__dict__.keys())
    def __hash__(self):
        return hash(tuple(sorted(self.__dict__.items())))
class Agent(Entity): pass
class Key(Entity): pass
class Door(Entity): pass
class Goal(Entity): pass
class Wall(Entity): pass
class Box(Entity): pass
class Ball(Entity): pass
class Lava(Entity): pass
def get_entities_by_name(entities, name):
    return [ entity for entity in entities if entity.name == name ]
def get_entities_by_position(entities, x, y):
    return [ entity for entity in entities if entity.x == x and entity.y ==
y ]
def reward_func(state, action, next_state):
    """
    Args:
        state: the state of the environment
        action: the action to be executed
        next_state: the next state of the environment
    Returns:
        reward: the reward of the action
        done: whether the episode is done
    """
    reward = 0.0  # initialise reward as 0.0 for all actions
    done = False # initialise done as False for all actions
    # extract the agent from the current and next state
    agent = get_entities_by_name(state, 'Agent')[0]
    next_agent = get_entities_by_name(next_state, 'Agent')[0]
    # If the agent picks up the green box in the next state, the reward is
1.0 and the episode is done
    if next_agent.carrying and isinstance(next_agent.carrying, Box) and
next_agent.carrying.color == 'green':
        reward = 1.0
        done = True
    return reward, done

```
```

```
Now, you have entered a new environment. It shows a mission "pick up the
yellow box". Do you know what this mission means and how to implement it in
a reward function? Analyze the behaviors of the reward function case by case
. In what situations will it return a positive reward or not? In what
situations will it return done=True or not? Why? Please implement the code
following the template in the sample
code. You must implement the ' reward_func' function as the main function to
 be called by the environment. The code needs to be directly runnable on (
mission, state, action, next_state) and return (reward, done) in python.
```

## F.6 Refine to satisfy optimism under uncertainty

It asks LLMs to think why the goal cannot be achieved for a mission from an initial state using
the provided transition and reward functions. We also tell LLMs the valid action space and the
measurement for achieving the goal ($r > 0 \land d = 1$).

```
You are a robot exploring in an object-centric environment. Your goal is to
model the logic of the world in python. You have tried it before and came up
 with one partially correct solution. However, it is not perfect. The code
can model the logic for some experiences but failed to model the logic to
achieve the goal in another environment. You need to improve your code so
that the agent can achieve the objective as specified by the mission from
the given initial state as well as still modelling the original logic. The
new code should still follow the same template. The ' transition ' function
needs to be directly runnable on (state, action) and return the next state
in python. The ' reward_func ' function needs to be directly runnable on (
state, action, next_state) and return (reward, done) in python.

Here is the partially correct solution you came up with:

'''

class Entity:
    def __init__(self, x, y, **kwargs):
        self.name = self.__class__.__name__
        self.x = x
        self.y = y
        for key, value in kwargs.items():
            setattr(self, key, value)
    def __repr__(self):
        attr = ', '.join(f'{key}={value}' for key, value in self.__dict__.
items() if key not in ('name', 'x', 'y'))
        if attr: return f"{self.name}({self.x}, {self.y}, {attr})"
        else: return f"{self.name}({self.x}, {self.y})"
    def __eq__(self, other):
        return all(getattr(self, key) == getattr(other, key, None) for key
in self.__dict__.keys())
    def __hash__(self):
        return hash(tuple(sorted(self.__dict__.items())))
class Agent(Entity): pass
class Key(Entity): pass
class Door(Entity): pass
class Goal(Entity): pass
class Wall(Entity): pass
class Box(Entity): pass
class Ball(Entity): pass
class Lava(Entity): pass
import copy
```

```python
def transition(state, action):
    """
    Args:
        state: the state of the environment
        action: the action to be executed
    Returns:
        next_state: the next state of the environment
    """
    # We'll make a deep copy of the state.
    # This is because we don't want to change the original state.
    next_state = copy.deepcopy(state)
    agent = get_entities_by_name(next_state, 'Agent')[0]
    if action == 'turn left':
        if agent.direction == (-1, 0):
            agent.direction = (0, 1)
        elif agent.direction == (0, 1):
            agent.direction = (1, 0)
        elif agent.direction == (1, 0):
            agent.direction = (0, -1)
        else:  # if agent.direction == (0, -1)
            agent.direction = (-1, 0)
    elif action == 'turn right':
        if agent.direction == (-1, 0):
            agent.direction = (0, -1)
        elif agent.direction == (0, -1):
            agent.direction = (1, 0)
        elif agent.direction == (1, 0):
            agent.direction = (0, 1)
        else:  # if agent.direction == (0, 1)
            agent.direction = (-1, 0)
    elif action == 'toggle':
        # We assume that the agent have the ability to toggle, regardless of
 what is in front of him because the experiences provided do not dictate
otherwise.
        pass
    elif action == 'nothing':
        pass
    return next_state
def get_entities_by_name(entities, name):
    return [ entity for entity in entities if entity.name == name ]
def get_entities_by_position(entities, x, y):
    return [ entity for entity in entities if entity.x == x and entity.y ==
y ]
def reward_func(state, action, next_state):
    """
    Args:
        state: the state of the environment
        action: the action to be executed
        next_state: the next state of the environment
    Returns:
        reward: the reward of the action
        done: whether the episode is done
    """
    # Create sets of entities for easier comparison and access
    state_set = set(state)
    next_state_set = set(next_state)
    # Get agent's position in both states
    agent = [e for e in state_set if isinstance(e, Agent)][0]
    agent_next = [e for e in next_state_set if isinstance(e, Agent)][0]
    # Done condition
    on_goal = any(isinstance(entity, Goal) for entity in
get_entities_by_position(next_state, agent_next.x, agent_next.y))
    done = on_goal
```

```
    # Reward calculation
    if state_set == next_state_set:
        # If state didn't change -> 'nothing', 'toggle' when not carrying a
key or 'drop' when carrying nothing happened
        reward = 0.0
    elif action == 'turn left' or action == 'turn right':
        # If direction of agent changes -> 'turn left', 'turn right'
happened
        reward = 0.0
    else:
        # In other cases, no reward. Can be modified when other scenarios
are applied.
        reward = 0.0
    return reward, done

'''
```

However, the code failed to achieve the goal/objective as specified by the
mission "use the key to open the door and then get to the goal" from the
following initial
state:

'''

```
Wall(0, 0) ;    Wall(1, 0) ;    Wall(2, 0) ;    Wall(3, 0) ;    Wall(4, 0) ;
Wall(0, 1) ;    empty ;         Wall(2, 1) ;    empty ;         Wall(4, 1) ;
Wall(0, 2) ;    Agent(1, 2, direction=(0, 1), carrying=None) ;  Door(2, 2,
color=yellow, state=locked) ;        empty ;         Wall(4, 2) ;
Wall(0, 3) ;    Key(1, 3, color=yellow) ;       Wall(2, 3) ;    Goal(3, 3) ;
    Wall(4, 3) ;
Wall(0, 4) ;    Wall(1, 4) ;    Wall(2, 4) ;    Wall(3, 4) ;    Wall(4, 4) ;
```

'''

The measurement for achieving the goal/objective is as follows:

'''

```
def criterion(state, mission, action, next_state, reward, done,):
    return reward > 0 and done
```

'''

The valid actions are {'turn right', 'nothing', 'move forward', 'turn left',
'toggle', 'drop', 'pick up'}.

Do you know why the mission cannot be achieved from the given initial state
with the world model as implemented in the code? What subgoals does the
agent need to achieve in order to achieve the final goal as specified by the
 mission? Can the agent achieve those subgoals using the world model as
implemented in the code? If not, what is missing or wrong? How can you
improve the code to achieve the goal/objective as specified by the mission
from the given initial state? Please improve the code as analyzed before so
that the mission can be achieved from the given initial
state. Please implement the code following the template. Feel free to
implement any helper functions you need. You can also implement the logic
for difference actions in different helper functions. However, you must
implement the ' transition ' function and the ' reward_func ' function as
the main functions to be called by the environment. The ' transition '
function needs to be directly runnable on (state, action) and return the
next state in python. The ' reward_func ' function needs to be directly
runnable on (state, action, next_state) and return (reward, done) in python.
 The new code, by themselves, should be complete, compilable, and runnable.

