# OpenReview forum: "WorldCoder, a Model-Based LLM Agent: Building World Models by Writing Code and Interacting with the Environment"
_NeurIPS.cc/2024/Conference — NeurIPS 2024 poster_

### Official Review · Reviewer_x859 · 2024-07-05

**Soundness:** 2
**Presentation:** 3
**Contribution:** 2
**Rating:** 4
**Confidence:** 4

**Summary:**

The authors propose a model-based agent by building a Python program to represent the world model through interacting with the environment.

The authors define logical constraints to explain the interactions and to achieve optimism in the face of (world model) uncertainty.

The authors prove the polynomial sample complexity, and conduct empirical study on Sokoban, Minigrid and AlfWorld.

**Strengths:**

+ a model-based agent
+ learn a world model through interactions with the environment
+ define logical constraints to explain the interactions and to achieve optimism in the face of (world model) uncertainty
+ prove the polynomial sample complexity
+ conduct empirical study on Sokoban, Minigrid and AlfWorld

**Weaknesses:**

- rely on an imperfect LLM and an imperfect method for program synthesis, and there is likely no way to achieve perfection
- it is not clear the effect of exponential search space on the polynomial sample complexity
- it is not clear if the evaluation is fair: 1) are the compared algorithms the best? 2) how about comparing problems deep RL is good at?

**Questions:**

The proposed method relies on an imperfect LLM and an imperfect method for program synthesis, how to handle errors due to such imperfection? Is there a way to achieve perfect program synthesis? If the answer is no, then the learned world model is not guaranteed to be correct.
This is a fundamental issue.

The paper is about learning a world model.
Is there a way to measure the quality of the learned model, e.g., comparing with the true model, rather than relying on performance study in the paper?
There should be such a perfect model for Sokoban and Minigrid.


What if integrating the proposed approach, in particular, \phi_1 & \phi_2 to deep RL, if feasible, maybe together with an LLM for language understanding?


How about MuZero?


How about some applications (deep) RL is good at, e.g, chess?



Line 90-91: “preferring an optimistic one guarantees that the agent can at least start making progress toward the goal, even if it later has to update its beliefs because they turned out to be too optimistic.”

Is this correct? Consider the shortest path problem. At some node, if an agent selects a neighbour / node, the agent may never achieve the optimal solution.


Under Line 97
Conditions \phi_1 fit data and \phi_2 optimization
What if there are noisy data points?
What if data points are contradicts with each other?

The meaning of condition \phi_2 is not clear, e.g., what is the meaning of \forall i \in [l]?
It is desirable to explain the conditions in plain English, and explain all notations.


Line 110
"Before ever receiving reward, \phi_2 forces the agent to invent a reward function."
Why and how to "invent a reward function"?
Why should it work?


Line 127
"we can learn in time polynomial w.r.t. the diameter (a property of the true MDP) and the logical dimensionality (a property of the model space)"
Theorem 2.4.
How about the diameter and the logical dimensionality for problems like chess and go?
Will they grow exponentially as the search space grows exponentially?


Line 146
"This retrieval strategy assumes similar goals have similar reward structures"
This may not be true. And how to define similarity?

Line 138
"without further learning from environmental interactions."
Line 149
"which update Rˆ(c) based on interactions with the environment"
These two statements contradict each other.

**Limitations:**

Yes.

---

> ### Author Rebuttal · Authors · 2024-08-05
>
> Thank you for the thoughtful review. Please see below for our responses.
>
> > What if integrating the proposed approach, in particular, \phi_1 & \phi_2 to deep RL, if feasible, maybe together with an LLM for language understanding?
>
> Interesting idea! One way we've been thinking about this is to frame the integration as hierarchical RL: have conventional deep RL handle low level control on small timescales, and our method act as a high level controller.
>
> > rely on an imperfect LLM and an imperfect method for program synthesis, and there is likely no way to achieve perfection
> > Is there a way to measure the quality of the learned model, e.g., comparing with the true model, rather than relying on performance study in the paper? There should be such a perfect model for Sokoban and Minigrid.
>
> We learn a perfect model on Sokoban and sometimes learn imperfect models on Minigrid which nonetheless achieve high reward, because perfectly modeling the transition function is not necessary to solve all tasks, especially when not all dynamics need be used to achieve the goal.
>
> But the real world is not as clean and tidy as videogames, so applying WorldCoder to something like a robot would require probabilistic programs, a direction we mention in the paper, and which we are excited to explore. We hope that by getting this paper out we can pursue these next steps. This relates to another comment:
>
> > Under Line 97 Conditions \phi_1 fit data and \phi_2 optimization What if there are noisy data points? What if data points are contradicts with each other?
>
> We currently consider deterministic environments without noise, but as line 311 of the paper says, we are pursuing future work that overcomes this limitation.
>
> > it is not clear the effect of exponential search space on the polynomial sample complexity
>
> We prove good sample complexity but not good computational complexity: in the worst case both planning and program synthesis are exponential time. We rely on the LLM to act as a good heuristic guide to make program synthesis not exponential in practice.
>
>
> > it is not clear if the evaluation is fair: 1) are the compared algorithms the best? 2) how about comparing problems deep RL is good at?
>
> We're happy to discuss any specific baselines you'd like performed, but we think any deep RL approaches are going to require many thousands or even millions of episodes, although they likely will asymptote to 100% success.
>
> The purpose of our work is not to show that Deep RL is always inferior, and in fact there are many problems that it can handle that our method would flop on. In the limitations section we mention that we require low-dimensional symbolic states and a deterministic environment, and comparing on any such problems would make deep RL look better. But given that our method is new and relatively unusual, we wanted to study interesting problems where it has advantages.
>
> > How about MuZero?
>
> We think it would perform very similarly to Dreamer_V3, which we compare against.
>
> > “preferring an optimistic [world model] guarantees that the agent can at least start making progress toward the goal"... Is this correct? Consider the shortest path problem. At some node, if an agent selects a neighbour / node, the agent may never achieve the optimal solution.
>
> We assume an episodic MDP, so even if the agent gets stuck on the current episode, it always has another try on the next episode---and if it was *too* optimistic, then it's guaranteed to not make the same mistake again, under our algorithm.
>
> > "Before ever receiving reward, \phi_2 forces the agent to invent a reward function." Why and how to "invent a reward function"? Why should it work?
>
> With no reward function, or a reward function that always returns zero, it is impossible to satisfy $\phi_2$, because that optimism constraint says that it should be possible to achieve positive reward from the initial state. In order to satisfy that constraint, there must exist a reward function such that a planner predicts reward can actually be achieved, using the current learned transition function. So the LLM is free to invent a reward function, subject to constraints in $\phi_2$
>
> > How about the diameter and the logical dimensionality for problems like chess and go? Will they grow exponentially as the search space grows exponentially?
>
> The diameter is at most twice the maximum length of a game. The logical dimensionality is approximately the log of the number of semantically distinct transition functions the language model can produce consistent with the replay buffer.
>
> > "assumes similar goals have similar reward structures" This may not be true. And how to define similarity?
>
> Indeed, this assumption is merely a heuristic. For example, if the agent has learned that the goal "pick up the ball" gives a reward of 10 for picking up the ball, then this assumption would say "pick up the balloon" should also give a reward of 10 for picking up a balloon. Similarity is implicitly defined by the LLM's in-context learning.
>
> Last, about the perceived contradiction between these statements:
> > L138: "After learning how the world works, our agent should be able to receive natural language instructions and begin following them immediately without further learning from environmental interactions"
>
> > L149: "If the LLM makes a mistake in predicting the reward function, then the agent can recover by subsequent rounds of program synthesis, which update $R(c)$ based on interactions with the environment."
>
> These statements say that the agent should be able to receive a new command and conjecture a plausible reward function, but also modify that reward function if it turns out to be incorrect, based on experience it collects later on down the line.
>
> > what is the meaning of $\forall i \in [l]$
>
> It means all integers $i$ where $1\leq i\leq l$. We'll revise to clarify this and other notation.
>
> Thanks for your input and please let us know if you have further questions.

---

> > ### Comment · Reviewer_x859 · 2024-08-10
> >
> > Thanks for the detailed explanation.
> >
> > The following question was not fully addressed yet:
> > The proposed method relies on an imperfect LLM and an imperfect method for program synthesis, how to handle errors due to such imperfection? Is there a way to achieve perfect program synthesis? If the answer is no, then the learned world model is not guaranteed to be correct. This is a fundamental issue.
> >
> > To evaluate the correctness of a piece of code, like pass@k, in the current popular approach, like HumanEval, if a solution passes several test cases, then it is a pass.
> > This is problematic: Testing only can not guarantee the correctness of a piece of code.  So it is not a reliable evaluation. Formal verification is required.
> >
> > The proposed method may learn a perfect model on Sokoban or achieve good experimental performance. However, without the guarantee of correctness of code / world model, we may not know what may happen.
> >
> >
> > An additional question.
> > Is the proposed world model learned with a symbolic approach equivalent to a transition model?  Is it better than a transition model? If equivalent, how about learning a transition model? Will learning a transition model simpler? Moreover, is the world model tabular? It seems so, from the definition of T and R at Line 78. If so, there is an issue how to do approximation for large problems.

---

> ### Author Response · Authors · 2024-08-10
>
> > The proposed method relies on an imperfect LLM and an imperfect method for program synthesis... the learned world model is not guaranteed to be correct... Formal verification is required... The proposed method may... achieve good experimental performance. However, without the guarantee of correctness of code / world model, we may not know what may happen.
>
> This is a fundamental issue *for model-based RL and machine learning broadly*, especially when neural networks are involved. However, representing knowledge as a program opens up unique opportunities for formal methods (see [1,2] for some of our favorite examples in RL+Programs), which we'd be excited to explore in the future. We'll revise to mention these factors, but our opinion is this criticism is not unique to our work and orthogonal to our contribution: An identical critique can be levied against neural network world models. Our method has the advantage that, in principle, there is hope of deploying formal methods in the future, owing to the symbolic nature of its learned knowledge.
>
> (If you're curious, we have ideas about how to formally verify safety properties of world models, which is an open problem, because past work [1-2] instead verifies policies. We can continue the discussion in that direction if you'd like. Might be interesting to get your feedback.)
>
> > Is the proposed world model learned with a symbolic approach equivalent to a transition model?
>
> The world model consists of a transition function $T:S\times A\to S$ and reward model $R:S\times A\times S\to \mathbb{R}$.
> The former is a transition model.
>
> > Moreover, is the world model tabular? It seems so
>
> The world model is not tabular. It does not memorize every possible case in a large table. The world model is a program which can generalize over states and actions.
>
> [1] An Inductive Synthesis Framework for Verifiable Reinforcement Learning. Zhu, Xiaog, Magill, Jagannathan. PLDI 2019.
>
> [2] Verifiable Reinforcement Learning via Policy Extraction. Bastani, Pu, Solar-Lezama. NeurIPS 2018.

---

> > ### Comment · Reviewer_x859 · 2024-08-10
> >
> > Thanks for your explanation.
> >
> > It is about if an LLM is reliably used.
> > LLMs are not fully reliable, as you also mention in Introduction.
> >
> > Your approach is based on program synthesis, which is based on an unreliable LLM, i.e., you can not guarantee the correctness of the program generated by the LLM.
> >
> > This is a fundamental issue.
> >
> > It is an issue with the recent LLM-based method, for most papers.
> > Exceptions are FunSearch and AlphaGeometry, which have reliable verifiers in the loop.
> >
> > It is not about neural networks.
> > For large scale problems, function approximation is inevitable, and neural networks are an option.
> >
> > AlphaGo and MuZero use neural networks for function approximation, but they do not suffer such an issue, because they have reliable feedback, i.e., game scores, for convergence.
> >
> > It is not about verification of RL or the world model, which are desirable though.
> > It is about correctness guarantee of the program generated, maybe by formal verification.

---

> > > ### Author Response · Authors · 2024-08-11
> > >
> > > Thanks for your continued engagement, which is helping understand the issue. We'd like to revised the paper by adding the following paragraph to the discussion, which discusses two ways in which formal methods could be used to check the correctness of a world model. Crucially there must be an externally provided logical specification of what the agent/world model should/shouldn't do, but there are different choices for what that specification should say and different ways of checking against the learned model. What do you think of this?
> > >
> > > > We prove WorldCoder converges to a *good enough* world model when granted enough compute and enough examples, meaning that the world model is sufficient to achieve high reward in the provided environments. However, we do not prove that the world model correctly predicts *all possible* (state, action) transitions, because certain transitions might not be relevant to optimal behavior within the provided environments. Although this deficiency is generically true for all model-based RL we are aware of, the fact that we represent the model as a program could open up unique opportunities for deploying formal methods to prove guarantees about all possible (state, actions). For example, given formula $\varphi$ characterizing unsafe behavior (e.g. in a temporal logic), we could use formal methods to try and prove that the agent acting optimally w.r.t. $(\hat{T},\hat{R})$ entails $\neg\varphi$, for all possible initial states. As another example, given a formal specification of the transition function dynamics $\Phi$, we could simply check if $\hat{T}\models \Phi $. Either way though, formal guarantees require an externally-provided logical specification (e.g. $\varphi$ or $\Phi$), which is absent from the standard RL setup.

---

> > > > ### Comment · Reviewer_x859 · 2024-08-11
> > > >
> > > > Thanks for your update.
> > > >
> > > > My point is:
> > > > All current LLMs are not imperfect, and your work is based on an imperfect LLM.
> > > >
> > > > Even there may be a verification step, current techniques, including LLMs, can not guarantee to fix the code to perfection.
> > > >
> > > > I believe your proof assumes the correctness of the code generated, or you do not consider such a factor in the proof.
> > > >
> > > > There does not seem to be a way for any LLM to improve to such a level, that it can generate a perfect piece of code, even given a perfect specification, which may not be available actually, not mention with an ambiguous natural language description. Not for code repair either.
> > > >
> > > > I understand that improving LLMs may be out of scope for you. But your work is based on imperfect LLMs.
> > > >
> > > > This is an issue not only to your submission, but to the LLM-based method.

---

> ### Author Response · Authors · 2024-08-12
>
> Thanks for continuing to engage with us. We think there's a misunderstanding we'd like to clarify.
>
> In the large-compute limit, our approach *does* converge to a program that perfectly fits the training examples. This is because the LLM always puts nonzero probability on at least one such program (at nonzero temperature), and so if code repair runs arbitrarily many times, it will perfectly fit the training examples. In practice we limit the compute budget, so might terminate before finding an optimal program, but the same is true for the other approaches you mention (FunSearch; AlphaGeometry). Our main theorem we prove only holds in the asymptotic compute regime, because it assumes we can perfectly fit the training examples. The theorem is nontrivial because it is not fundamentally about fitting the training examples, but instead about RL sample-complexity, i.e. achieving high reward by successfully exploring the environment in relatively few steps.
>
> We hope that this large-compute limit clarifies the relationship between imperfect LLMs and our method, as well as FunSearch and AlphaGeometry.
>
> > I understand that improving LLMs may be out of scope for you. But your work is based on imperfect LLMs. This is an issue not only to your submission, but to the LLM-based method.
>
> Indeed, we agree that it's not unique to our submission and likely out of scope. We just want to emphasize that our method, like others such as FunSearch and AlphaGeometry, are only asymptotically *complete* in the large-compute limit---although they are always *sound* because they can decline to output a program if it fails to verify/satisfy the examples: So we always have soundness but only get completeness asymptotically. Distinguishing soundness and completeness is important here.

---

> > ### Comment · Reviewer_x859 · 2024-08-13
> >
> > Thanks for your explanation.
> >
> > Yes, we are not on the same page.
> >
> > Let me summarize my point briefly.
> >
> > 1. LLMs are imperfect.
> >
> > 2. Solutions generated / repaired by imperfect LLMs can not guarantee correctness.
> >
> > 3. To remedy it, a verifier is required. AlphaGeometry has such a verifier.
> >
> > 4. Your method is based on an imperfect LLM, and does not have a verifier.
> >
> > It is not about scale of compute, or convergence.
> > It is about a fundamental flaw with reliance on imperfect LLMs.

---

### Official Review · Reviewer_WssY · 2024-07-07

**Soundness:** 3
**Presentation:** 3
**Contribution:** 3
**Rating:** 7
**Confidence:** 4

**Summary:**

The paper presents WorldCoder, a system for learning world models through program synthesis with an LLM. The framing is model-based reinforcement learning, where the agent interacts with the environment and collects data that is used to learn the model. WorldCoder uses a general-purpose language to synthesize the model and a test-and-debug cycle to improve the accuracy of the model.

The paper also uses the idea of optimism in the face of uncertainty of the learned model. This is achieved by preferring models that can lead to positive rewards. Even if the positive reward is not achieved in practice, this forces the agent to explore the space. Empirical results demonstrate the potential of the approach.

**Strengths:**

The paper makes contributions to the important problem of learning world models for sequential decision making. The solution is also interesting as it uses program synthesis with a general-purpose language; most previous work relies on domain-specific languages.

The paper is very well written and easy to follow. It provides examples where needed, and reading it was a pleasure.

The results are quite strong compared to deep RL methods. This is particularly interesting given that the gain is coming from learning a model of the environment. Model-based RL can be quite finicky, and the results this paper produces are outstanding.

The comparisons of versions of the system with optimism and without optimism are also good and important to have published.

Given that WorldCoder uses an LLM, it is also nice to see that the cost of using the model is much lower than other systems used for planning. Overall, the use of LLM to learn world models makes perfect sense and is promising.

**Weaknesses:**

My concern with the paper regards the validity of the experiments given how the LLM is trained. GPT-4, the model used in the experiments, certainly was trained on the code implementing the models of the domains used in the experiments. How much of what we see is from the general knowledge of the LLM versus memorizing the implementation of the models used in the experiments?

The paragraph about React (Section 3) attempts to deal with this issue, but it misses the point. It argues that having knowledge of Sokoban isn't enough to solve problems. This is true, but it is not what WorldCoder could use. WorldCoder would benefit from simply knowing the code of the models, search will do the rest. Having knowledge of the model is way more likely than having useful knowledge of how to solve Sokoban puzzles. Sokoban is computationally hard. It is difficult to imagine how one would be able to code Sokoban knowledge to benefit systems like React. On the other hand, it is very easy to code knowledge that would benefit WorldCoder: implementations of the models would suffice.

Data contamination is the only reason why I am giving a "poor" score to soundness.

**Questions:**

Do you agree that the data contamination issue is not explained by the paragraph that starts with "Almost surely" in Section 3?

How much of the code of the models for the three domains is available online?

How would WorldCoder perform on a domain that is different from everything that is out there?

**Limitations:**

The paper discusses most of the limitations. The limitation that is missing is the data contamination I mentioned in the weaknesses section.

---

> ### Author Rebuttal · Authors · 2024-08-05
>
> Thank you for the thoughtful review. We really think we can improve the paper to address your main points. Please see below.
>
> > GPT-4, the model used in the experiments, certainly was trained on the code implementing the models of the domains used in the experiments. How much of what we see is from the general knowledge of the LLM versus memorizing the implementation of the models used in the experiments?
>
> This is an excellent point and it deserves careful consideration:
>
> 1. In the attached PDF (general response) we show what happens when the agent is trained on a new variation of Sokoban we designed this week, and which therefore could not be in pretraining data. The new variant has warp gates/teleportation portals allowing the agent to instantly move between a pair of locations. The logic of the teleport gate is subtle, because it is deactivated whenever one of the portals is blocked by a box. The agent nonetheless learns how to use the warp gates, and solves problems faster by exploiting them to take shortcuts, as shown in the PDF.
>
>
> 2. When solving AlfWorld, it makes many edits to its world knowledge to correct problems with its prior understanding of this environment (Fig 5 bottom), for example learning that only one thing can be picked up at a time, or that objects cannot be removed from closed receptacles. When solving Sokoban, it has to make analogous fixes, such as understanding that boxes can block the movement of other boxes, or that there is a per-step reward of -0.1. If it were simply retrieving the answer from training data, we would not expect these mistakes. Instead, it seems to be retrieving general knowledge about how videogames work, and making lots of small changes to the transition function based on the details of the data in the replay buffer.
>
> > How much of the code of the models for the three domains is available online?
>
> 1. AlfWorld is publicly available as a zip file containing PDDL configs. It is therefore unlikely that GPT4 was trained on its implementation, both because the raw data is zipped, and because it is not in a common source code file extension (PDDL looks like a weird lisp with an unusual file extension).
> 2. Minigrid (released 2022) and Sokoban are probably in the pretraining data. Our new Sokoban result is not in the pretraining data.
>
> We will revise the paper to put more emphasis on the AlfWorld results and the new Sokoban results, because those are the ones that we are the most sure are not affected by data contamination.
>
> > The paragraph about React (Section 3) attempts to deal with this issue, but it misses the point
>
> Upon reading your argument, we agree with you that the ReAct baseline does not address these concerns. The revision will swap that discussion for the discussion above.
>
> > How would WorldCoder perform on a domain that is different from everything that is out there?
>
> We've designed a new Sokoban domain to answer that question. The best way of answering that question would be to design completely new games for the agent that have never been on the internet, or barring that, to use domains like AlfWorld whose environment implementation is very unlikely to have been trained on.
>
> > Data contamination is the only reason why I am giving a "poor" score to soundness
>
> We hope that you can reconsider your score in light of the new experiment and the above discussion.
>
> Thank you for the unusually helpful review. Please let us know if you have any further questions.

---

> > ### Comment · Reviewer_WssY · 2024-08-12
> >
> > Thank you for carrying out experiments on a different version of Sokoban and explaining the availability of AlfWorld in the training data. I have increased my score accordingly.

---

### Official Review · Reviewer_9H9M · 2024-07-12

**Soundness:** 3
**Presentation:** 4
**Contribution:** 2
**Rating:** 6
**Confidence:** 3

**Summary:**

- This work deals with the transfer of text-based LLMs to the domain of agents, enabling LLMs to build a (world) model of an environment.
- The paper introduces the algorithm 'WorldCoder', that creates and refines Python programs for approximation of state transition and reward function
  based on sampled data from the environment.
- Hereby, the program search and refinement is conditioned on two constraints:
 (1. fit data) no state transition inaccuracies (batch) w.r.t. the observed data
 (2. optimism) the reward function should give a reward atleast for one state, enabling exploration.

**Strengths:**

- The paper is technically sound and seems generally reproducible.
- Good results, high sample effiency and task generalization for experiments, but limited domain with some assumptions.
- Topic: Bridging the gap between LLMs and the agent domain by building world models with code via env interactions seems an interesting area to explore.
  Researchers are probably interested in this idea.

**Weaknesses:**

- Equation 1,2 are hard to parse.

- Task generalization achieved by prior knowledge: Env generalization with uncertain/incomplete prior knowledge of LLMs is not properly addressed.
  E.g., the dynamics behavior of grid worlds is obviously known by GPT-4 and therefore we can expect generalized dynamics in the transition function code.

- Scalability is not properly addressed. (See question)

- Since Program Synthesis via Refinement (Equation 3 (REx)) is not developed in this work and only adapted to the problem, the paper's contribution boils
  down to prompt-engineering with output constraints, then reiterating over output content with additional prompts for debugging.

**Questions:**

- Scalability is not properly addressed. E.g. looking at the prompts in Appendix for MiniGrid: the states are encoded in a rigerous manner with
  a lot of designed structure and tokens. How does the performance behave for large state space encodings. Is the LLM still able to create adequate transition functions?

- Experiments on grids or highly abstracted env states: Is this an analogy to Kahnemann's System 2 or a current limitation?

**Limitations:**

- The authors do not address input data integrity: The method relies on perfect(?) input sample quality. If not, then the LLM can infer wrong rules for
  the transition function. Since each rules is abstract, the program can also generalise wrong. Thus, the robustness for low quality data is not addressed.

- The generated Python programs can only model highly abstracted MDPs, since the input samples are a text-based, tokenized abstractions.

- Experiments on grids or highly abstracted env states => How to ensure hiqh quality input data in the general case (e.g. pixels)?

---

> ### Author Rebuttal · Authors · 2024-08-05
>
> Thank you for the thoughtful review, and for your support. Please see below for our responses.
>
> > Task generalization achieved by prior knowledge: Env generalization with uncertain/incomplete prior knowledge of LLMs is not properly addressed. E.g., the dynamics behavior of grid worlds is obviously known by GPT-4
>
> The system leverages its prior knowledge, but is not bound by it. Here are two examples, including new results in the attached PDF:
> 1. In the attached PDF we show what happens when the agent is trained on a new variation of Sokoban we designed that has warp gates/teleportation portals allowing the agent to instantly move between a pair of locations, as long as one of them is not blocked by a box. The agent learns how to use the warp gates and can solve problems faster by exploiting them, as shown.
> 2. When solving AlfWorld, it makes many edits to its world knowledge to correct problems with its prior (Fig 5 bottom), for example learning that only one thing can be picked up at a time, or that objects cannot be removed from closed receptacles.
>
> > the paper's contribution boils down to prompt-engineering with output constraints
>
> The contribution includes a new logical/formal framing of optimism under uncertainty, its theoretical analysis, and empirically showing this framing works in practice and helps accelerate learning. We further contribute an LLM agent with better big-$\mathcal{O}$ complexity in terms of amortized # API calls per action, a consequence of being model-based instead of model-free.
>
> > Scalability is not properly addressed. E.g. looking at the prompts in Appendix for MiniGrid: the states are encoded in a rigerous manner with a lot of designed structure and tokens. How does the performance behave for large state space encodings. Is the LLM still able to create adequate transition functions?
>
> We scale to complicated symbolic environments: AlfWorld has a very large state space with typically around 60 objects and around 20 predicates, for a states space size of $2^{60\times 20}$. Appendix E lists an example learned world model for this setting. We do not consider nonsymbolic environments, although it is a next step that we are actively working on.
>
> > Experiments on grids or highly abstracted env states: Is this an analogy to Kahnemann's System 2 or a current limitation?
>
> It is indeed analogous to System-2 abstract reasoning, or task planning in robotics more specifically
>
> > The authors do not address input data integrity: The method relies on perfect(?) input sample quality… The generated Python programs can only model highly abstracted MDPs, since the input samples are a text-based, tokenized abstractions… Experiments on grids or highly abstracted env states => How to ensure hiqh quality input data in the general case (e.g. pixels)?
>
> We are working on extensions that learn probabilistic programs (to handle noise in the data), and which use multimodal models to define the basic symbols/tokens (so we can learn from e.g. pixels), which would allow handling, for example, robotics environments. We consider these extensions sufficiently different that they should go into a different paper, while WorldCoder_v1 is more concerned with the fundamentals of the problem and its conceptual/mathematical solution.

---

> > ### Comment · Reviewer_9H9M · 2024-08-12
> >
> > Dear Authors,
> >
> > thanks for your clarification. I raise my rating by one point, because of
> > emphasized contributions (a bit more than raw prompt engineering), the additional experiment (with unknown dynamics in Sokoban context) and
> > AlfWorld explanation. However, since AlfWorld env is a bit less known (at least to me), it should get a better introduction?
> > AlfWorld state encoding/tokenization not added in the Appendix?
> >
> > I think it is an interesting approach with great potential.
> > As LLMs increasingly get more sophisticated, this method could also scale with them.
> >
> > Best regards,
> > Reviewer 9H9M

---

### Official Review · Reviewer_NhBf · 2024-07-13

**Soundness:** 3
**Presentation:** 3
**Contribution:** 3
**Rating:** 7
**Confidence:** 3

**Summary:**

This paper introduces a novel model-based agent for sequential decision-making. A structured world model and goal-conditioned reward function are created by an LLM and refined as new information or task changes come in. This is based on logical constraints. For selecting the right candidate world model (ie object-oriented transition function), the constraint is that the world model explain all observed transitions up to now. For selecting the right goal-conditioned reward function, the constraint is that the reward function be optimistic, which means that it should assume that there is at least one trajectory reaching the goal.

After proving a theoretical guarantee on the maximum number of actions to reach the goal, the authors move on to empirical evaluation. There are 3 evaluation environments: sokoban, minigrid and alfworld. On sokoban, a conceptually simple but hard grid-world puzzle, the proposed agent largely outperforms ReAct, an LLM-agent baseline in terms of score, as well as DreamerV3 and PPO in sample efficiency (by orders of magnitude). On minigrid, a sparse-reward gridworld for language instruction-following agents, the authors successfully demonstrate transfer between world models and rewards from one environment to the next in a curriculum, and demonstrate that when there is no curriculum variants without optimistic rewards fail to solve the tasks. The third environment is AlfWorld, a text-based environment modelling a kitchen environment with several rooms and objects, some of which have affordances such as being cut, opened, turned on, etc... The authors showcase the scalability of their method by demonstrating that it can learn a faithful world model of the domain and successfully solve all tasks in AlfWorld very rapidly.

**Strengths:**

* The method is original and interesting. While it bears similarity with previous work using programs to represent transition functions, it is the first to have used LLMs to infer program-based world models written in a general-purpose programming language;
* The paper is quite clear and reads well;
* The work is adequately contextualized and relevant references are given along with a comprehensive account of the paper's position in relation to them;
* Experiments support the claims and contributions stated by the authors.
* I could see other researchers using methods inspired by worldcoder on their own domains; when the approach transfers, it is a massive gain in efficiency compared to other types of agents;

**Weaknesses:**

* The method remains slightly complex and nontrivial to implement from the paper alone, making easily available code an important part of the re-usability of the method;
* The method lacks generality, needing a hand-defined curriculum in minigrid and a handcrafted language-based reward bonus on AlfWorld;
* Compared with state of the art LLM-agents (like Voyager), the environments considered here (especially AlfWorld) are pretty simple. Could WorldCoder perform well on ScienceWorld, an environment with more interesting dynamics and objectives? (breed plants, measure boiling temperatures, etc). Could it, in principle, scale to the text-based version of MineCraft used in Voyager?
* I think the paper lacks solid baselines in the LLM-agent case. ReAct is a very basic agent, and a more principled baseline for tasks like sokoban would be Tree of Thoughts (ToT), dfs version, that also performs planning. It could be plausible that ToT performs well, in which case the compute efficiency claims still hold.
* The paper lacks baselines in minigrid and alfworld.

**Questions:**

* First sentence of the abstract is not grammatically correct, it seems?
* Could a version of WorldCoder work for hard linguistic/reasoning tasks (see limitation below)? [SWE bench](https://www.swebench.com/), [theorem-proving in Lean](https://leandojo.org/), etc?
* What are typical logical dimensionalities of the environments you consider?
* How do you ensure the self-repair program synthesis step does not break previously satisfied conditions for the transition function or the reward function?
* The hand-designed curriculum limits generality of the approach, but could you leverage [Automatic Curriculum Learning](https://arxiv.org/abs/2003.04664) to create environments in the zone of proximal development?
* line 289 “A disadvantage is that policies cannot readily generalize to new goals” except if policies are parametrized by the goal, which programming languages allow you to do: go_to_object(goal) would certainly transfer to many different objects.

**Limitations:**

* Limitation: this agent cannot perform the complex linguistic tasks that humans do, since it is limited by the planner to operate in domains with a restricted number of actions (and that excludes the space of all possible sentences). The comparison to ReAct should mention this.
* See above for curriculum and hand-defined reward;
* Otherwise the authors acknowledge some additional limitations, such as the lack of modelling of stochasticity in their world model or the need for the observations to come pre-segmented in objects (or the world to be organized in recognizable, distinct objects at all).

---

> ### Author Rebuttal · Authors · 2024-08-05
>
> Thank you for the comprehensive review and for being supportive of the paper. Please below for answers to your questions.
>
> > The method lacks generality, needing a hand-defined curriculum in minigrid and a handcrafted language-based reward bonus on AlfWorld;
>
> We think this might be a misunderstanding:
> 1. The method doesn't need a curriculum (see Fig 4C), but can take advantage of good curriculum if one is provided (Fig 4B)
> 2. The reward bonus does not contain any AlfWorld-specific features: It relies on generic text features. It can apply to other domains where the objects and actions have names in natural language.
>
> > the environments considered here… are pretty simple
>
> We started with simple environments because our pilot experiments showed that new basic principles needed to be developed (optimism under uncertainty applied to programs). We are excited to explore richer application domains such as those you mention, and we hope that by getting this work published we can soon begin that follow-up work, which requires addressing the limitations outlined in the submission.
>
> Please note that in the general response and in the attached PDF we have designed a more complicated Sokoban environment---with warp gates/teleportation portals--and tested our approach on it as part of responding to a different reviewer.
>
> > a more principled baseline for tasks like sokoban would be Tree of Thoughts (ToT)
>
> We will work on this and hopefully have something to share during the discussion period next week. However the asymptotic analysis would be the same as ReAct, with cost growing linearly as a function of the number of episodes.
>
> > The method remains slightly complex and nontrivial to implement... easily available code an important part
>
> Code and data will be released upon publication.
>
> > if policies are parametrized by the goal, which programming languages allow you to do: go_to_object(goal) would certainly transfer to many different objects
>
> Good point! We will mention that.
>
> > The paper lacks baselines in minigrid and alfworld
>
> Please see Appendix Fig 6 for minigrid Deep RL baselines. We agree though that more baselines on those domains would help the paper, but we also believe strongly that deep RL baselines would always (1) eventually solve every task but (2) take millions of episodes. On the other hand LLM agent baselines would (1) rapidly achieve sub-100% performance but (2) never asymptote to 100%, due to hallucinations and (3) have API cost linear in the number of episodes. We observed these characteristics for Sokoban (deep RL/ReAct) and Minigrid (deep RL), and given the high cost of running experiments, felt it was low priority to run further simulations.
>
> > Could a version of WorldCoder work for hard linguistic/reasoning tasks
>
> Thanks for suggesting these directions. WorldCoder could be used for the tasks you mentioned if the world model can delegate parts of its computation to a "fuzzy" reasoning module like an LLM, similar to ViperGPT. This would be a fascinating direction to explore.
>
> > How do you ensure the self-repair program synthesis step does not break previously satisfied conditions
>
> We keep all the old experience data in a replay buffer to ensure that we do not break the world model when updating it for new data.
>
>
> Thanks again for your support and please let us know if you have further questions.

---

> > ### Comment · Reviewer_NhBf · 2024-08-08
> >
> > I thank you for taking the time to respond to my comments and questions in detail, and I understand better some parts of the paper as a result. I am convinced that WorldCoder is a very strong method under the class of environment you consider (symbolic and deterministic) and am looking forward to follow up work in stochastic, fuzzy, or linguistic environments.

---

### Author Rebuttal · Authors · 2024-08-05

Thank you everyone for the helpful input! Although there is a separate response to each review, we wanted to also include a global response to highlight a new experimental result this week performed to help better understand how the system adapts to new environments that are different from what it is seen in pretraining data.

For this new environment we added warp gates/teleportation portals to Sokoban that allow the agent to instantly transport itself. The logic is subtle because the gates become deactivated if one of them is blocked by a box. In the attached PDF we show an example of how the final trained agent solves problems using the teleporters in order to solve puzzles more quickly by taking shortcuts. Learning takes 2-3 episodes to learn the nuances of warp gates, depending on the random seed.

Please see attachment.

---

### Decision · Program_Chairs · 2024-09-25

**Decision:**

Accept (poster)

**Comment:**

Overall most reviewers found the work to be an interesting and novel way of approaching sequential decision making with LLMs.

A number of potential limitations and presentation issues were raised by the reviewers with most being well addressed in the rebuttals.

The authors are encouraged to leverage the reviewers' feedback on the presentation and results to improve the final camera ready paper.